# SE(3)-bi-equivariant Transformers
# for Point Cloud Assembly

**Ziming Wang, Rebecka Jörnsten**
Department of Mathematical Sciences
University of Gothenburg and Chalmers University of Technology
`zimingwa@chalmers.se, jornsten@chalmers.se`

## Abstract

Given a pair of point clouds, the goal of assembly is to recover a rigid transformation that aligns one point cloud to the other. This task is challenging because the point clouds may be non-overlapped, and they may have arbitrary initial positions. To address these difficulties, we propose a method, called $SE(3)$-bi-equivariant transformer (BITR), based on the $SE(3)$-bi-equivariance prior of the task: it guarantees that when the inputs are rigidly perturbed, the output will transform accordingly. Due to its equivariance property, BITR can not only handle non-overlapped PCs, but also guarantee robustness against initial positions. Specifically, BITR first extracts features of the inputs using a novel $SE(3) \times SE(3)$-transformer, and then projects the learned feature to group $SE(3)$ as the output. Moreover, we theoretically show that swap and scale equivariances can be incorporated into BITR, thus it further guarantees stable performance under scaling and swapping the inputs. We experimentally show the effectiveness of BITR in practical tasks.

## 1 Introduction

Point cloud (PC) assembly is a fundamental machine learning task with a wide range of applications such as biology [12], archeology [36], robotics [28, 20] and computer vision [23]. As shown in Fig. 1, given a pair of 3-D PCs representing two shapes, *i.e.*, a source and a reference PC, the goal of assembly is to find a rigid transformation, so that the transformed source PC is aligned to the reference PC. This task is challenging because the input PCs may have random initial positions that are far from the optimum, and may be non-overlapped, *e.g.*, due to occlusion or erosion of the object.

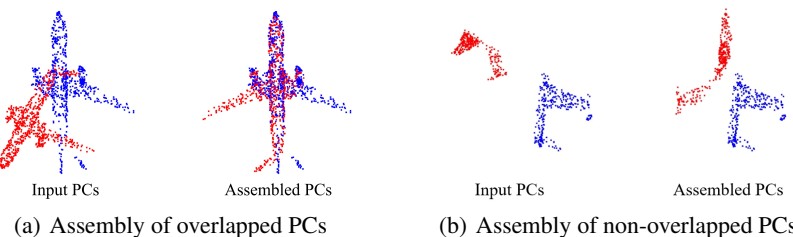

| Input PCs | Assembled PCs | | Input PCs | Assembled PCs |
| --- | --- | --- | --- | --- |
| (a) Assembly of overlapped PCs | | | (b) Assembly of non-overlapped PCs | |

Figure 1: Two examples of PC assembly. Given a pair of PCs, the proposed method BITR transforms the source PC (red) to align the reference PC (blue). The input PCs may be overlapped (a) or non-overlapped (b).

Most of existing assembly methods are correspondence-based [2, 23, 4]: they use the fact that when the input PCs are aligned, the points corresponding to the same physical position should be close to

38th Conference on Neural Information Processing Systems (NeurIPS 2024).

each other. For example, in Fig. 1(a), the points at the head of the airplane in the source and reference PCs should be close in the aligned PC. Specifically, these methods first estimate the correspondence between PCs based on feature similarity or distance, and then compute the transformation by matching the estimated corresponding point pairs. As a result, these methods generally have difficulty handling PCs with no correspondence, *i.e.*, non-overlapped PCs, such as Fig. 1(b). In addition, they are often sensitive to the initial positions of PCs.

To address these difficulties, we propose a method, called $SE(3)$-bi-equivariant transformer (BITR), based on the $SE(3)$-bi-equivariance prior of the task: when the input PCs are perturbed by rigid transformations, the output should transform accordingly. A formal definition of $SE(3)$-bi-equivariance can be found in Def. 3.1. Our motivation for using the $SE(3)$-bi-equivariance prior is threefold: First, the strong training guidance provided by symmetry priors is known to lead to large performance gain and high data efficiency. For example, networks with a translation-equivariance prior, *i.e.*, convolutional neural networks (CNNs), are known to excel at image segmentation [18]. Thus, $SE(3)$-bi-equivariance prior should lead to similar practical benefits in PC assembly tasks. Second, $SE(3)$-bi-equivariant methods are theoretically guaranteed to be "global", *i.e.*, their performances are independent of the initial positions. Third, the $SE(3)$-bi-equivariance prior does not rely on correspondence, *i.e.*, it can be used to handle PCs with no correspondence.

Specifically, the proposed BITR is an end-to-end trainable model consisting of two steps: it first extracts $SO(3) \times SO(3)$-equivariant features from the input PCs, and then obtains a rigid transformation by projecting the features into $SE(3)$. For the first step, we define a $SE(3) \times SE(3)$-transformer acting on the 6-D merged PC by extending the $SE(3)$-transformer [11]; For the second step, we use a *SVD*-type projection inspired by Arun's method [2]. In addition, we theoretically show that scale-equivariance and swap-equivariance can be incorporated into BITR via weight constraining techniques, which further guarantees that the performance is not influenced by scaling or swapping inputs. An illustration of BITR is presented in Fig. 2.

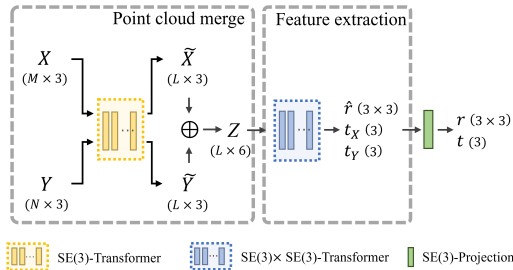

Figure 2: An overview of BITR. The input 3-D PCs $X$ and $Y$ are first merged into a 6-D PC $Z$ by concatenating the extracted key points $\tilde{X}$ and $\tilde{Y}$. Then, $Z$ is fed into a $SE(3) \times SE(3)$-transformer to obtain equivariant features $\hat{r}$, $t_X$ and $t_Y$. These features are finally projected to $SE(3)$ as the output.

In summary, the contribution of this work is as follows:

- We present a $SE(3)$-bi-equivariant PC assembly method, called BITR. [1] BITR can assemble PCs without correspondence, and guarantees stable performance with arbitrary initial positions. In addition, the $SE(3) \times SE(3)$-transformer used in BITR is the first $SE(3) \times SE(3)$-equivariant steerable network to the best of our knowledge.

- Theoretically, we show that scale and swap equivariance can be incorporated in to BITR by weight-constraining, thus it guarantees stable performance under scaling and swapping the inputs.

- We show experimentally that BITR can effectively assemble PCs in practical tasks.

## 2   Related works

A special case of PC assembly is PC registration, where the correspondence between input PCs is assumed to exist. A seminal work in this task was conducted by [2], which provided a closed-form solution to the problem with known correspondence. To handle PCs with unknown correspondence, most of the subsequent works extend [2] by first estimating the correspondence by comparing distances [4], or features [25, 23, 41] of the PCs, and then aligning the PCs by aligning the estimated corresponding points. Notably, to obtain $SE(3)$-bi-equivariance, $SO(3)$-invariant features [42, 10, 43] have been investigated for correspondence estimation. However, since these methods require a sufficient number of correspondences, they have difficulty handling PCs where the correspondence does not exist. In addition, they often have difficulty handling PCs with large initial errors [45].

---

The proposed BITR is related to the existing registration methods because it can be seen as a generalization of Arun's method [2]. However, in contrast to these methods, BITR is correspondence-free, *i.e.*, it is capable of handling PCs with no correspondence. In addition, it theoretically guarantees stable performance under arbitrary initial position.

Recently, some works have been devoted to a new PC assembly task, called fragment reassembly, whose goal is to reconstruct a complete shape from two fragments. Unlike registration task, this task generally does not assume the existence of correspondence. [7] first studied this task, and they addressed it as a pose estimation problem, where the pose of each fragment relative to a canonical pose is estimated via a regression model. [38] further improved this method by considering the $SE(3)$-equivariance of each fragment. In addition, [31] proposed a simulated dataset for this task. In contrast to these methods, the proposed BITR does not rely on the canonical pose, *i.e.*, it directly estimates the relative pose. As a result, BITR is conceptually simpler and it can handle the shape whose canonical pose is unknown.

Another related research direction is equivariant networks. Due to their ability to incorporate 3D rotation and translation symmetry priors, these networks have been extensively used in modelling 3D data [29, 9, 37, 17, 13], and recently they have been used for robotic manipulation task [27, 28, 40]. In particular, [34] proposed a tensor field network (TFN) for PC processing, and $SE(3)$-transformer [11] further improved TFN by introducing the attention mechanism. On the other hand, the theory of equivariant networks was developed in [16, 5]. BITR follows this line of research because the $SE(3) \times SE(3)$-transformer used in BITR is a direct generalization of $SE(3)$-transformer [11], and it is the first $SE(3) \times SE(3)$-equivariant steerable network to the best of our knowledge.

## 3 Preliminaries

This section briefly reviews Arun's method and the concept of equivariance, which will be used in BITR.

### 3.1 Group representation and equivariance

Given a group $G$, its representation is a group homomorphism $\rho : G \to GL(V)$, where $V$ is a linear space. When $G$ is the 3D rotation group $SO(3)$, it is convenient to consider its irreps (irreducible orthogonal representation) $\rho_p : SO(3) \to GL(V_p)$, where $p \in \mathbb{N}$ is the degree of the irreps, and $dim(V_p) = 2p + 1$. For $r \in G$, $\rho_p(r) \in \mathbb{R}^{(2p+1)\times(2p+1)}$ is known as the Wigner-D matrix. For example, $\rho_0(r) = 1$ for all $r \in SO(3)$; $\rho_1(r) \in \mathbb{R}^{3\times3}$ is the rotation matrix of $r$. More details can be found in [5] and the references therein.

In this work, we focus on the group $G$ of two independent rotations, *i.e.*, $G = SO(3) \times SO(3)$, where $\times$ represents the direct product. Similar to $SO(3)$, we also consider the irreps of $G$. A useful fact is that all irreps of $G$ can be written as the combinations of the irreps of $SO(3)$: the degree-$(p, q)$ irreps of $G$ is $\rho_{p,q} = \rho_p \otimes \rho_q : SO(3) \times SO(3) \to GL(V_p \otimes V_q)$, where $p, q \in \mathbb{N}$, $\rho_p$ and $\rho_q$ are irreps of $SO(3)$, and $\otimes$ is tensor product (Kronecker product for matrix). For example, $\rho_{0,0}(r_1 \times r_2) = 1 \in \mathbb{R}$; $\rho_{1,0}(r_1 \times r_2) \in \mathbb{R}^{3\times3}$ is the rotation matrix of $r_1$; $\rho_{1,1}(r_1 \times r_2) = \rho_1(r_1) \otimes \rho_1(r_2) \in \mathbb{R}^{9\times9}$ is the Kronecker product of the rotation matrices of $r_1$ and $r_2$.

Given two representations $\rho : G \to GL(V)$ and $\tau : G \to GL(W)$, a map $\Phi : V \to W$ satisfying $\Phi(\rho(g)x) = \tau(g)\Phi(x)$ for all $g \in G$ and $x \in V$ is called $G$-equivariant. When $\Phi$ is parametrized by a neural network, we call $\Phi$ an equivariant neural network, and we call the feature extracted by $\Phi$ equivariant feature. Specifically, a degree-$p$ equivariant feature transforms according to $\rho_p$ under the action of $SO(3)$, and a degree-$(p, q)$ equivariant feature transforms according to $\rho_p \otimes \rho_q$ under the action of $SO(3) \times SO(3)$. For simpler notations, we omit the representation homomorphism $\rho$, *i.e.*, we write $r$ instead of $\rho(r)$, when $\rho$ is clear from the text.

### 3.2 Arun's method

Consider a PC assembly problem *with known one-to-one correspondence*: Let $Y = \{y_u\}_{u=1}^{N} \subseteq \mathbb{R}^3$ and $X = \{x_u\}_{u=1}^{N} \subseteq \mathbb{R}^3$ be a pair of PCs consisting of $N$ points, and let $\{(x_u, y_u)\}_{u=1}^{N}$ be their corresponding point pairs. What is the optimal rigid transformation that aligns $X$ to $Y$?

[2] provided a closed-form solution to this problem. It claims that the optimal solution $g = (r, t) \in SE(3)$ in terms of mean square errors is

$$r = SVD(\bar{r}) \quad \text{and} \quad t = \boldsymbol{m}(Y) - r\boldsymbol{m}(X) \tag{1}$$

where

$$\bar{r} = \sum_i \bar{y}_i \bar{x}_i^T \tag{2}$$

is the correlation matrix, $\boldsymbol{m}(\cdot)$ represents the mean value, $\bar{x}_i = x_i - \boldsymbol{m}(X)$ and $\bar{y}_i = y_i - \boldsymbol{m}(Y)$ represent the centralized points, and $SVD(\cdot)$ represents the singular value decomposition projection. The definition of SVD projection can be found in Def. C.5.

Arun's solution enjoys rich equivariance properties. Formally, we have the following proposition:

**Definition 3.1.** Consider a map $\Phi : \mathbb{S} \times \mathbb{S} \to SE(3)$ where $\mathbb{S}$ is the set of 3D PCs. Given $X, Y \in \mathbb{S}$, let $g = \Phi(X, Y)$.

- $\Phi$ is $SE(3)$-bi-equivariant if $\Phi(g_1 X, g_2 Y) = g_2 g g_1^{-1}, \quad \forall g_1, g_2 \in SE(3)$.

- $\Phi$ is swap-equivariant if $\Phi(Y, X) = g^{-1}$.

- $\Phi$ is scale-equivariant if $\Phi(cX, cY) = (r, ct), \ \forall c \in \mathbb{R}_+$.

**Proposition 3.2.** *Under a mild assumption (C.2), Arun's algorithm (1) is $SE(3)$-bi-equivariant, swap-equivariant and scale-equivariant.*

In other words, Arun's method guarantees to perform consistently 1) with arbitrary rigid perturbations on $X$ and $Y$, *i.e.*, it is global, 2) if $X$ and $Y$ are swapped (aligning $Y$ to the fixed $X$ or aligning $X$ to the fixed $Y$), and 3) in arbitrary scale. Details of Prop. 3.2 can be found at Appx. C.1.

We regard Arun's method as a prototype of $SE(3)$-bi-equivariant PC assembly methods: it first extracts a degree-$(1, 1)$ $SO(3) \times SO(3)$-equivariant translation-invariant feature, *i.e.*, the correlation matrix $\bar{r}$ (2), and then obtains an output $g \in SE(3)$ using a *SVD*-based projection. This observation immediately leads to more general $SE(3)$-bi-equivariant methods where the handcrafted feature $\bar{r}$ is replaced by the more expressive learned equivariant features, thus, the correspondence is no longer necessary. We will develop this idea in the proposed BITR in the next section, and further show that the scale and swap equivariance of Arun's method can also be inherited.

## 4 $SE(3)$-bi-equivariant transformer

This section presents the details of the proposed BITR. BITR follows the same principle as Arun's method [2]: it first extracts $SO(3) \times SO(3)$-equivariant features as a generalization of the correlation matrix $\bar{r}$ (2), and then projects the features to $SE(3)$ similarly to (1). Specifically, we first propose a $SE(3) \times SE(3)$-transformer for feature extraction in Sec. 4.2. Since this transformer is defined on 6-D space, *i.e.*, it does not directly handle the given 3-D PCs, it relies on a pre-processing step described in Sec. 4.3, where the input 3-D PCs are merged into a 6-D PC. Finally, the Arun-type $SE(3)$-projection is presented in Sec. 4.4. An overview of BITR is presented in Fig. 2.

### 4.1 Problem formulation

Let $Y = \{y_v\}_{v=1}^N \subseteq \mathbb{R}^3$ and $X = \{x_u\}_{u=1}^M \subseteq \mathbb{R}^3$ be the PCs sampled from the reference and source shape respectively. The goal of assembly is to find a rigid transformation $g \in SE(3)$, so that the transformed PC $gX = \{rx_i + t\}_{i=1}^M$ is aligned to $Y$. Note that we do not assume that $X$ and $Y$ are overlapped, *i.e.*, we do not assume the existence of corresponding point pairs.

### 4.2 $SE(3) \times SE(3)$-transformer

To learn $SO(3) \times SO(3)$-equivariant translation-invariant features generalizing $\bar{r}$ (1), this subsection proposes a $SE(3) \times SE(3)$-transformer as a generalization of $SE(3)$-transformer [11]. We present a brief introduction to $SE(3)$-transformer [11] in Appx. A for completeness.

According to the theories developed in [5], to define a $SE(3) \times SE(3)$-equivariant transformer, we first need to define the feature map of a transformer layer as a tensor field, and specify the action

of $SE(3) \times SE(3)$ on the field. Since $SE(3) \times SE(3)$ can be decomposed as $(T(3) \times T(3)) \rtimes (SO(3) \times SO(3))$ where $T(3)$ is the 3-D translation group, the tensor field should be defined in the 6-D Euclidean space $\mathbb{R}^6 \cong T(3) \times T(3)$, and the features attached to each location should be $SO(3) \times SO(3)$-equivariant and $T(3) \times T(3)$-invariant. Formally, we define the tensor field as

$$f(z) = \sum_{u=1}^{L} f_u \delta(z - z_u) \tag{3}$$

where $Z = \{z_u\}_{u=1}^{L} \subseteq \mathbb{R}^6$ is a 6-D PC, $\delta$ is the Dirac delta function, $f_u = \oplus_{p,q} f_u^{p,q}$ is the feature attached to $z_u$, where $f_u^{p,q}$ is the degree-$(p,q)$ equivariant component. We then specify the action of $SE(3) \times SE(3)$ on the base space $\mathbb{R}^6$ as

$$(g_1 \times g_2)(z) = (g_1 z^1) \oplus (g_2 z^2) \quad \forall z \in \mathbb{R}^6 \tag{4}$$

where $z = z^1 \oplus z^2$, $z^1, z^2 \in \mathbb{R}^3$ are the first and last three components of $z$, $\oplus$ represents direct sum (concatenate), and $g_i = (r_i, t_i) \in SE(3)$ for $i = 1, 2$. Thus, the action of $SE(3) \times SE(3)$ on the degree-$(p,q)$ component of $f$ is

$$\left((g_1 \times g_2) f^{p,q}\right)(z) = \left(\rho_{p,q}(r_1 \times r_2)\right) f^{p,q} \left((g_1 \times g_2)^{-1}(z)\right).$$

With the above preparations, we can now define a $SE(3) \times SE(3)$-transformer layer in a message passing formulation similar to $SE(3)$-transformer:

$$f_{\text{out}}^{\boldsymbol{o}}(z_u) = \underbrace{W^{\boldsymbol{o}} F_{\text{in}}^{\boldsymbol{o}}(z_u)}_{\text{self-interaction}} + \sum_{\substack{\boldsymbol{i} \\ v \in KNN(u) \backslash \{u\}}} \underbrace{\alpha_{uv}}_{\text{attention}} \underbrace{\mathbf{V}_{uv}^{\boldsymbol{o},\boldsymbol{i}}}_{\text{message}}. \tag{5}$$

Here, we use notations $\boldsymbol{o} = (o_1, o_2)$ and $\boldsymbol{i} = (i_1, i_2)$ for simplicity. For example, $f^{\boldsymbol{o}}$ represents the degree-$\boldsymbol{o}$ feature $f^{o_1,o_2}$. $F^{\boldsymbol{o}}(z_u) \in \mathbb{R}^{c \times (2o_1+1)(2o_2+1)}$ is the collection of all degree-$\boldsymbol{o}$ features at $z_u$, where $c$ is the number of channels of the degree-$\boldsymbol{o}$ features, and $W^{\boldsymbol{o}} \in \mathbb{R}^{1 \times c}$ is a learnable parameter for self-interaction. $KNN(\cdot)$ represents the $k$ nearest neighborhood, and attention $\alpha_{uv}$ is computed according to

$$\alpha_{uv} = \frac{\exp\left(\mathbf{Q}_u^\top \mathbf{K}_{uv}\right)}{\sum_{v' \in KNN(u) \backslash \{u\}} \exp\left(\mathbf{Q}_u^\top \mathbf{K}_{uv'}\right)}, \tag{6}$$

where $\mathbf{Q}$, $\mathbf{K}$ and $\mathbf{V}$ are known as the query, key and value respectively. They are defined as

$$\mathbf{Q}_u = \bigoplus_{\boldsymbol{o}} W_Q^{\boldsymbol{o}} F_{\text{in}}^{\boldsymbol{o}}(z_u), \mathbf{K}_{uv} = \bigoplus_{\boldsymbol{o}} \sum_{\boldsymbol{i}} \mathcal{W}_K^{\boldsymbol{o},\boldsymbol{i}}(z_{vu}) f_{\text{in}}^{\boldsymbol{i}}(z_v), \mathbf{V}_{uv}^{\boldsymbol{o},\boldsymbol{i}} = \mathcal{W}_V^{\boldsymbol{o},\boldsymbol{i}}(z_{vu}) f_{\text{in}}^{\boldsymbol{i}}(z_v) \tag{7}$$

where $z_{vu} = z_v - z_u$, $W_Q^{\boldsymbol{i}}$ is a learnable parameter for $\mathbf{Q}$, and the convolutional kernel $\mathcal{W}^{\boldsymbol{o},\boldsymbol{i}}(z)$ in $\mathbf{V}$ and $\mathbf{K}$ both take the form of

$$vec(\mathcal{W}^{\boldsymbol{o},\boldsymbol{i}}(z)) = \sum_{J_1=|o_1-i_1|}^{o_1+i_1} \sum_{J_2=|o_2-i_2|}^{o_2+i_2} \left( \underbrace{\varphi_{J_1,J_2}^{\boldsymbol{o},\boldsymbol{i}}(\|z^1\|, \|z^2\|)}_{\text{radial component}} \underbrace{\mathcal{Q}_{J_1,J_2}^{\boldsymbol{o},\boldsymbol{i}} Y_{J_1}(\frac{z^1}{\|z^1\|}) \otimes Y_{J_2}(\frac{z^2}{\|z^2\|})}_{\text{angular component}} \right), \tag{8}$$

where the learnable radial component $\varphi_{J_1,J_2}^{\boldsymbol{o},\boldsymbol{i}} : \mathbb{R} \times \mathbb{R} \to \mathbb{R}$ is parametrized by a neural network, the non-learnable angular component is determined by the 2-nd order Clebsch-Gordan constant $\mathcal{Q}$ and the spherical harmonics $Y_J : \mathbb{R}^3 \to \mathbb{R}^{2J+1}$, and $vec(\cdot)$ is the vectorize function reshaping a matrix to a vector. Formulation (8) is derived in Appx. B.

Note that the kernel (8) is the main difference between a $SE(3)$-transformer layer and a $SE(3) \times SE(3)$-transformer layer. In the special case where only $SE(3)$-equivariance is considered, *i.e.*, all features are of degree $(p, 0)$ (or $(0, q)$), a $SE(3) \times SE(3)$-transformer layer becomes a $SE(3)$-transformer layer.

Finally, we adopt the equivariant Relu (Elu) layer similar to [9] as the point-wise non-linear layer in our network. Given an input degree-$\boldsymbol{i}$ feature $F^{\boldsymbol{i}}$ with $c$ channels, an Elu layer is defined as

$$Elu(F^{\boldsymbol{i}}) = \begin{cases} F_\mu & \langle F_\mu, F_\nu \rangle \geqslant 0 \\ F_\mu - \langle F_\mu, \frac{F_\nu}{\|F_\nu\|} \rangle \frac{F_\nu}{\|F_\nu\|} & \text{otherwise,} \end{cases} \tag{9}$$

where $F_\mu = W_\mu^{\boldsymbol{i}} F^{\boldsymbol{i}}$ and $F_\nu = W_\nu^{\boldsymbol{i}} F^{\boldsymbol{i}}$. $W_\mu, W_\nu \in \mathbb{R}^{c \times c}$ are learnable weights and $\|\cdot\|$ is the channel-wise vector norm. Note that when $\boldsymbol{i} = (1, 0)$ or $\boldsymbol{i} = (0, 1)$, our definition (9) becomes the same as [9]. By interleaving transformer layers and Elu layers, we can finally build a complete $SE(3) \times SE(3)$-equivariant transformer model.

### 4.3 Point cloud merge

To utilize the transformer model defined in Sec. 4.2, we need to construct a 6-D PC as its input. To this end, we first extract key points from the raw 3-D PCs, and then concatenate them to a 6-D PC to merge their information. Thus, the resulting 6-D PC is not only small in size but also contains the key information of the raw PCs pairs.

Formally, we extract $L$ ordered key points $\tilde{X} = \{\tilde{x}_u\}_{u=1}^{L}$ and $\tilde{Y} = \{\tilde{y}_u\}_{u=1}^{L}$ from $X$ and $Y$ respectively, and then obtain $Z = \{\tilde{x}_u \oplus \tilde{y}_u\}_{u=1}^{L}$. Note that we do not require $\tilde{X}$ ($\tilde{Y}$) to be a subset of $X$ ($Y$). Specifically, we represent the coordinates of the key points as a convex combination of the raw PCs:

$$\tilde{X} = SoftMax(F_X^0)X, \quad \tilde{Y} = SoftMax(F_Y^0)Y, \tag{10}$$

where $X \in \mathbb{R}^{M \times 3}$ and $Y \in \mathbb{R}^{N \times 3}$ represent the coordinates of $X$ and $Y$ respectively, and $SoftMax(\cdot)$ represents the row-wise softmax. $F_X^0 \in \mathbb{R}^{L \times M}$ and $F_Y^0 \in \mathbb{R}^{L \times N}$ are the weights of each point in $X$ and $Y$ respectively, and they are degree-0, *i.e.*, rotation-invariant, features computed by a shared $SE(3)$-transformer $\Phi_E$:

$$F_X^0 = \Phi_E(X), \quad F_Y^0 = \Phi_E(Y). \tag{11}$$

Furthermore, inspired by [39], we fuse the features of $X$ and $Y$ in $\Phi_E$ before the last layer, so that their information is merged more effectively, *i.e.*, the selection of $\tilde{X}$ or $\tilde{Y}$ depends on both $X$ and $Y$. Specifically, the fused features are

$$\begin{cases} f_{\text{out},X}(x_u) = f_{\text{in},X}^0(x_u) \oplus Pool_v\left(f_{\text{in},Y}^0(y_v)\right) \oplus f_{\text{in},X}^1(x_u) \\ f_{\text{out},Y}(y_v) = f_{\text{in},Y}^0(y_v) \oplus Pool_u\left(f_{\text{in},X}^0(x_u)\right) \oplus f_{\text{in},Y}^1(y_v), \end{cases} \tag{12}$$

where we only consider degree-0 and degree-1 features. $f_{\cdot,X}$ and $f_{\cdot,Y}$ represent the features of $X$ and $Y$, and *Pool* is the average pooling over the PC.

Note that $\tilde{X}$ and $\tilde{Y}$ obtained in Eqn. 10 are permutation invariant. For example, according to Eqn. 10, the $i$-th key point of $X$ is $\tilde{x}_i = \sum_j \mathcal{F}_{ij} X_j$, where $\mathcal{F}_{ij}$ is the $i$-th channel of $F_X^0$ at $x_j$ (after softmax normalization). When $X$ is permutated by $\sigma$, then the $i$-th key point can be written as $\tilde{x}_i' = \sum_{j'} \mathcal{F}_{ij'} X_{j'}$, where $j' = \sigma(j)$. It is easy to see that $\tilde{x}_i' = \tilde{x}_i$ because both $j$ and $j'$ iterate through $\{1, ..., M\}$ in the summation.

### 4.4 SE(3)-projection

We now obtain the final output by projecting the feature extracted by the $SE(3) \times SE(3)$-transformer to $SE(3)$. Formally, let $f$ be the output tensor field of the $SE(3) \times SE(3)$-transformer. We compute the final output $g = (r, t) \in SE(3)$ using an Arun-type projection as follows:

$$\begin{cases} r = SVD(\hat{r}) \\ t = (\boldsymbol{m}(\tilde{Y}) + t_Y) - r(\boldsymbol{m}(\tilde{X}) + t_X), \end{cases} \tag{13}$$

where $\hat{r} = unvec(\tilde{r}) \in \mathbb{R}^{3 \times 3}$, $t_X \in \mathbb{R}^3$ and $t_Y \in \mathbb{R}^3$ are equivariant features computed as

$$\tilde{r} = Pool_u(f_u^{1,1}), \ t_X = Pool_u(f_u^{1,0}), \ t_Y = Pool_u(f_u^{0,1}).$$

We note that projection (13) extends Arun's projection (1) in two aspects. First, although $\hat{r}$ in (13) and $\bar{r}$ (2) are both degree-$(1, 1)$ features, $\hat{r}$ is more flexible than $\bar{r}$ because $\hat{r}$ is a learned feature while $\bar{r}$ is handcrafted, and $\hat{r}$ is correspondence-free while $\bar{r}$ is correspondence-based. Second, projection (13) explicitly considers non-zero offsets $t_X$ and $t_Y$, which allow solutions where the centers of PCs do not match.

In summary, BITR computes the output $g$ for PCs $X$ and $Y$ according to

$$g = \Phi_P \circ \Phi_S(X, Y), \tag{14}$$

where $\Phi_S : \mathbb{S} \times \mathbb{S} \to \mathbb{F}$ is a $SE(3) \times SE(3)$-transformer (with the PC merge step), $\Phi_P : \mathbb{F} \to SE(3)$ represents projection (13), and $\mathbb{F}$ is the set of tensor field. We finish this section with a proposition that BITR is indeed $SE(3)$-bi-equivariant.

**Proposition 4.1.** *Under a mild assumption (C.2), BITR (14) is $SE(3)$-bi-equivariant.*

# 5 Swap-equivariance and scale-equivariance

This section seeks to incorporate swap and scale equivariances into the proposed BITR model. These two equivariances are discussed in Sec. 5.1 and Sec. 5.2 respectively.

## 5.1 Incorporating swap-equivariance

This subsection seeks to incorporate swap-equivariance to BITR, *i.e.*, to ensure that swapping $X$ and $Y$ has the correct influence on the output. To this end, we need to treat the group of swap as $\mathbb{Z}/2\mathbb{Z} = \{1, s\}$ where $s^2 = 1$, *i.e.*, $s$ represents the swap of $X$ and $Y$, and properly define the action of $\mathbb{Z}/2\mathbb{Z}$ on the learned features.

Formally, we define the action of $\mathbb{Z}/2\mathbb{Z}$ on field $f$ (3) as follows. We first define the action of $s$ on the base space $\mathbb{R}^6$ as swapping the coordinates of $\tilde{X}$ and $\tilde{Y}$: $s(z) = z^2 \oplus z^1$, where $z = z^1 \oplus z^2$, and $z^1, z^2 \in \mathbb{R}^3$ are the coordinates of $\tilde{X}$ and $\tilde{Y}$ respectively. Then we define the action of $s$ on feature $f$ as $(s(f))^{p,q}(z) = (f^{q,p}(s(z)))^T$, where we regard a degree-$(p,q)$ feature $f_u^{p,q}$ as a matrix of shape $\mathbb{R}^{(2p+1)\times(2q+1)}$ by abuse of notation, and $(\cdot)^T$ represents matrix transpose.

Intuitively, according to the above definition, degree-$(1,1)$, $(1,0)$ and $(0,1)$ features will become (the transpose of) degree-$(1,1)$, $(0,1)$ and $(1,0)$ features respectively under the action of $s$, *i.e.*, $\hat{r}$ will be transposed, $t_X$ and $t_Y$ will be swapped. This is exactly the transformation needed to ensure swap-equivariant outputs. We formally state this observation in the following proposition.

**Proposition 5.1.** *For a tensor field $f$ and a projection $\Phi_P$ (13), $\Phi_P(s(f)) = (\Phi_P(f))^{-1}$.*

Now the remaining problem is how to make a $SE(3) \times SE(3)$-transformer $\mathbb{Z}/2\mathbb{Z}$-equivariant. A natural solution is to force all layers in the $SE(3) \times SE(3)$-transformer to be $\mathbb{Z}/2\mathbb{Z}$-equivariant. The following proposition provides a concrete way to achieve this.

**Proposition 5.2.** *Let $\tilde{\cdot}$ represent the swap of index, e.g., if $\boldsymbol{o} = (o_1, o_2)$, then $\tilde{\boldsymbol{o}} = (o_2, o_1)$. 1) For a transformer layer (5), if the self-interaction weight satisfies $W^{\boldsymbol{o}} = W^{\tilde{\boldsymbol{o}}}$, the weight of query (7) satisfies $W_Q^{\boldsymbol{o}} = W_Q^{\tilde{\boldsymbol{o}}}$, and the radial function satisfies $\varphi_{J_1,J_2}^{\boldsymbol{i},\boldsymbol{o}}(\|z^1\|, \|z^2\|) = \varphi_{J_2,J_1}^{\tilde{\boldsymbol{i}},\tilde{\boldsymbol{o}}}(\|z^2\|, \|z^1\|)$ for all $\boldsymbol{i}, \boldsymbol{o}, J_1, J_2, z^1$ and $z^2$, then the transformer layer is $\mathbb{Z}/2\mathbb{Z}$-equivariant.*

*2) For an Elu layer (9), if $W_\nu^{\boldsymbol{i}} = W_\nu^{\tilde{\boldsymbol{i}}}$ and $W_\mu^{\boldsymbol{i}} = W_\mu^{\tilde{\boldsymbol{i}}}$ for all $\boldsymbol{i}$, then the Elu layer is $\mathbb{Z}/2\mathbb{Z}$-equivariant.*

More details, including the complete matching property (Prop. C.11), can be found in Appx. C.3.1.

## 5.2 Incorporating scale-equivariance

This subsection seeks to incorporate scale equivariance to BITR, *i.e.*, to ensure that when $X$ and $Y$ are multiplied by a scale constant $c \in \mathbb{R}_+$, the output result transforms correctly. To this end, we need to consider the scale group $(\mathbb{R}_+, \times)$, *i.e.*, the multiplicative group of $\mathbb{R}_+$, and properly define the $(\mathbb{R}_+, \times)$-equivariance of the learned feature. For simplicity, we abbreviate group $(\mathbb{R}_+, \times)$ as $\mathbb{R}_+$.

We now consider the action of $\mathbb{R}_+$ on field $f$ (3). We call $f$ a *degree-$p$ $\mathbb{R}_+$-equivariant field* ($p \in \mathbb{N}$) if it transforms as $(c(f))(z) = c^p f(c^{-1}z)$ under the action of $\mathbb{R}_+$, where $z \in \mathbb{R}^6$ and $c \in \mathbb{R}_+$. We immediately observe that degree-1 $\mathbb{R}_+$-equivariant features lead to scale-equivariant output. Intuitively, if $\tilde{r}$, $t_X$ and $t_Y$ are degree-1 $\mathbb{R}_+$-equivariant features, then they will become $c\tilde{r}$, $ct_X$ and $ct_Y$ under the action of $c$, and the projection step will cancel the scale of $\tilde{r}$ while keeping the scale of $t_X$ and $t_Y$, which is exactly the desirable results. Formally, we have the following proposition.

**Proposition 5.3.** *Let $\Phi_P$ be projection (13), $f$ be a degree-1 $\mathbb{R}_+$-equivariant tensor field, and $(r, t) = \Phi_P(f)$. We have $\Phi_P(c(f)) = (r, ct) \quad \forall c \in \mathbb{R}_+$.*

The remaining problem is how to ensure that a $SE(3) \times SE(3)$-transformer is $\mathbb{R}_+$-equivariant and its output is of degree-1, so that scaling the input can lead to the proper scaling of output. Here we provide a solution based on the following proposition.

**Definition 5.4.** *$\varphi : \mathbb{R} \times \mathbb{R} \to \mathbb{R}$ is a degree-$p$ function if $\varphi(cx, cy) = c^p \varphi(x, y)$ for all $c \in \mathbb{R}_+$.*

**Proposition 5.5.** *1) Denote $\varphi_K$ and $\varphi_V$ the radial functions used in $\mathbf{K}$ and $\mathbf{V}$ respectively. Let $\varphi_K$ be a degree-0 function, $f_{in}$ be a degree-0 $\mathbb{R}_+$-equivariant input field. For transformer layer (5), if $\varphi_V$*

*is a degree-1 function and the self-interaction weight $W = 0$, then the output field $f_{out}$ is degree-1 $\mathbb{R}_+$-equivariant; If $\varphi_V$ is a degree-0 function, then the output field $f_{out}$ is degree-0 $\mathbb{R}_+$-equivariant.*

*2) For Elu layer (9), if the input field is degree-$p$ $\mathbb{R}_+$-equivariant, then the output field is also degree-$p$ $\mathbb{R}_+$-equivariant.*

More discussions can be found in Appx. C.4.

## 6 Experiments and analysis

This section experimentally evaluates the proposed BITR. After describing the experiment settings in Sec. 6.1, we first present a simple example in Sec. 6.2 to highlight the equivariance of BITR. Then we evaluate BITR on assembling the shapes in ShapeNet [6], BB dataset [31], 7Scenes [32] and ASL [22] from Sec. 6.3.1 to Sec. 6.4. We finally apply BITR to visual manipulation tasks in Sec. 6.6.

### 6.1 Experiment settings

We extract $L = 32$ key points for each PC. The $SE(3)$-transformer and the $SE(3) \times SE(3)$-transformer both contain 2 layers with $c = 4$ channels. We consider $k = 24$ nearest neighborhoods for message passing. We only consider low degree equivariant features, *i.e.*, $p, q \in \{0, 1\}$ for efficiency. We train BITR using Adam optimizer [15] with learning rate $1e^{-4}$. We use the loss function $L = \|r^T r_{gt} - I\|_2^2 + \|t_{gt} - t\|_2^2$, where $(r, t)$ are the output transformation, $(r_{gt}, t_{gt})$ are the corresponding ground truth. We evaluate all methods by isotropic rotation and translation errors: $\Delta r = (180/\pi) accos \left( 1/2 \left( tr(rr_{gt}^T) - 1 \right) \right)$, and $\Delta t = \|t_{gt} - t\|$ where $tr(\cdot)$ is the trace of a matrix. We do not use random rotation and translation augmentations as [23]. More details are in Appx. D.1.

### 6.2 A proof-of-concept example

To demonstrate the equivariance property of BITR, we train BITR on the bunny shape [33]. We prepare the dataset similar to [41]: In each training iteration, we first construct the raw PC $S$ by uniformly sampling 2048 points from the bunny shape and adding 200 random outliers from $[-1, 1]^3$, then we obtain PCs $\{X_P, Y_P\}$ by dividing $S$ into two parts of ratio $(30\%, 70\%)$ using a random plane $P$. We train BITR to reconstruct $S$ using randomly rotated and translated $\{X_P, Y_P\}$. To construct the test set, we generate a new sample $\{X_{\tilde{P}}, Y_{\tilde{P}}\}$, and additionally construct 3 test samples by 1) swapping, 2) scaling (factor 2) and 3) randomly rigidly perturbing $\{X_{\tilde{P}}, Y_{\tilde{P}}\}$.

The assembly results of BITR on these four test samples are shown in Fig. 3. We observe that BITR performs equally well in all cases. Specifically, the differences between the rotation errors in these four cases are small (less than $1e^{-3}$). The results suggest that BITR is indeed robust against these three perturbations, which verifies its swap-equivariance, scale-equivariance and $SE(3)$-bi-equivariance. More experiments can be found in the appendix: a numerical verification of Def. 3.1 is presented in Appx. D.2, an ablation study of swap and scale equivariances are presented in Appx. D.3, and the verification of the complete-matching property C.11 is presented in Appx. D.4.

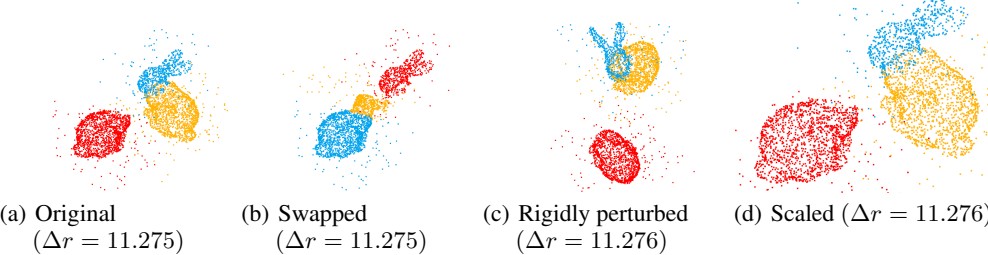

(a) Original     (b) Swapped     (c) Rigidly perturbed   (d) Scaled ($\Delta r = 11.276$)
 ($\Delta r = 11.275$)   ($\Delta r = 11.275$)   ($\Delta r = 11.276$)

Figure 3: The results of BITR on a test example (a), and the swapped (b), scaled (d) and rigidly perturbed (c) inputs. The red, yellow and blue colors represent the source, transformed source and reference PCs respectively.

## 6.3 Results on ShapeNet

### 6.3.1 Single shape assembly

In this experiment, we evaluate BITR on assembling PCs sampled from a single shape. When the inputs PCs are overlapped, this setting is generally known as PC registration. We construct a dataset similar to [41]: for a shape in the airplane class of ShapeNet [6], we obtain each of the input PCs by uniformly sampling $1024$ points from the shape, and keep ratio $s$ of the raw PC by cropping it using a random plane. We vary $s$ from $0.7$ to $0.3$. *Note the PCs may be non-overlapped when $s < 0.5$.*

We compare BITR against the state-of-the-art registration methods GEO [23] and ROI [43], and the state-of-the-art fragment reassembly methods NSM [7] and LEV [38]. For NSM and LEV, we additionally provide the canonical pose for each shape. Note that LEV and ROI are $SE(3)$-equivariant methods. For $s \geq 0.5$, we report the results of BITR fine-tuned by ICP [45] (BITR+ICP). Note that BITR+ICP is $SE(3)$-bi-equivariant and scale-equivariant, but not swap-equivariant.

We present the results in Fig. 4. We observe that the performance of all methods decrease as $s$ decreases. Meanwhile, BITR outperforms all baseline methods when $s$ is small ($s \leq 0.5$). On the other hand, when $s$ is large ($s > 0.5$), BITR performs worse than GEO, but it still outperforms other baselines. Nevertheless, since the results of BITR are sufficiently close to optimum ($\Delta r \leq 20$), the ICP refinement can lead to improved results that are close to GEO. More details can be found in Appx. D.5.

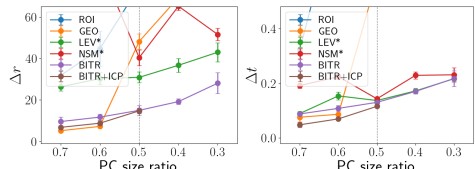

Figure 4: Assembly results on the airplane dataset. $*$ denotes methods which require the true canonical poses of the input PCs.

### 6.3.2 Inter-class assembly

To evaluate BITR on non-overlapped PCs, we extend the experiment in Sec. 6.3.1 to inter-class assembly. We train BITR to place a car shape on the right of motorbike shape, so that their directions are the same and their distance is $1$. We consider $s = 1.0$ and $0.7$. Note that this task is beyond the scope of registration methods, since the input PCs are non-overlapped. A result of BITR is shown in Fig. 5. More details can be found in Appx. D.6.

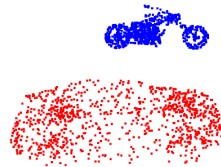

Figure 5: A result of BITR on assembling a motorbike and a car.

## 6.4 Results on fragment reassembly

This subsection evaluates BITR on a fragment reassembly task. We compare BITR against NSM [7], LEV [38] and DGL [44] on the 2-fragment WineBottle class of the BB dataset [31]. The data preprocessing step is described in Appx. D.7.

We test the trained BITR 3 times using different random samples, and report the mean and standard deviation of $(\Delta r, \Delta t)$ in Tab. 1. We observe that BITR outperforms all baseline methods: BITR achieves the lowest rotation errors, and its translation error is comparable to DGL, which is lower than other baselines by a large margin. We provide some qualitative comparisons in Appx. D.7.

Table 1: Reassembly results on 2-fragment WineBottle. We report the mean and std of the error of BITR.

|  | $\Delta r$ | $\Delta t$ |
|---|---|---|
| DGL | 101.3 | 0.09 |
| NSM | 101 | 0.18 |
| LEV | 98.3 | 0.17 |
| BITR (Ours) | 8.4 (0.9) | 0.07 (0.008) |

## 6.5 Results on real data

This subsection evaluates BITR on an indoor dataset 7Scenes [32] and the outdoor scenes in ASL dataset [22]. We present the results on 7Scenes in this section, and leave the results on ASL and some qualitative results to Appx. D.8.

For the 7Scenes dataset, we arbitrarily rotate and translate all frames, and train BITR to align all adjacent frames. We use the data from the first 6 scenes as the training set, and the data from 7-th

scene as the test set. To train BITR, we use a random clipping augmentation similar to Sec. 6.3.1: we keep ratio $s$ of each PCs by clipping them using a random plane, where $s$ is uniformly distributed in $[0.5, 1.0]$. We compare BITR against GEO [23], ROI [43], ICP [45] and OMN [39], where OMN is a recently proposed correspondence-free registration method.

The results on 7Scenes are reported in Tab. 2. We observe that BITR can produce results that are close to the optimum ($\Delta r \approx 25$) from a random initialization ($\Delta r \in U[0, 180]$), and extra refinements like ICP and OT can further improve the results ($\Delta r \approx 10$). This observation is consistent with that in Sec. 6.3.1. In particular, BITR with the OT refinement is comparable with GEO and ROI, which use highly complicated features specifically designed for registration tasks and an OT-like refinement process. On the other hand, ICP and OMN fails in this task due to their sensitiveness to initial positions.

Table 2: Results on 7Scenes. We report mean and std of $\Delta r$ and $\Delta t$.

|  | $\Delta r$ | $\Delta t$ |
|---|---|---|
| ICP | 73.2 (5.7) | 2.4 (0.2) |
| OMN | 129.02 (2.15) | 0.98 (0.06) |
| GEO | 9.2 (0.02) | 0.2 (0.08) |
| ROI | 9.0 (0.0) | 0.2 (0.0) |
| BITR (Ours) | 26.7 (0.0) | 0.8 (0.0) |
| BITR+ICP (Ours) | 11.1 (0.0) | 0.3 (0.0) |
| BITR+OT (Ours) | 9.5 (0.0) | 0.3 (0.0) |

## 6.6 Results on visual manipulation

This subsection investigates the potential of BITR in manipulation tasks. Following [26], we consider two tasks: mug-hanging and bowl-placing. For both tasks, $X$ represents an object grasped by a robotic arm, *i.e.*, a cup or a bowl, $Y$ represents the fixed environment with a target, *i.e.*, a stand or a plate, and we train BITR to align $X$ to $Y$, so that the cup can be hung to the stand, or the bowl can be placed on the plate.

Fig. 6 presents the results of BITR on bowl-placing. We observe that although BITR is not originally designed for manipulation tasks, it can place the bowl in a reasonable position relative to the plate. However, we also notice that BITR may produce unrealistic results, *e.g.*, the PCs may collide. Thus, post-processing steps [30] or extra regularizers [13] may be necessary in practical applications. More results and discussions can be found in Appx. D.9.

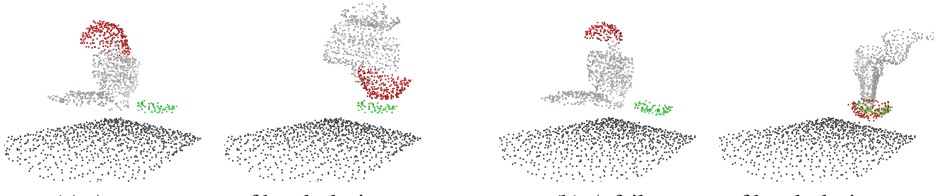

(a) A success case of bowl-placing                    (b) A failure case of bowl-placing

Figure 6: The results of BITR on bowl-placing. We present the input PCs (left panel) and the assembled results (right panel). BITR can generally place the bowl (red) on the plate (green) (a), but it sometimes produces unrealistic results where collision exists (b).

## 7 Conclusion

This work proposed a PC assembly method, called BITR. The most distinguished feature of BITR is that it is correspondence-free, $SE(3)$-bi-equivariant, scale-equivariant and swap-equivariant. We experimentally demonstrated the effectiveness of BITR.

BITR in its current form has two main limitations. First, BITR is computationally inefficient because each degree of feature is computed independently without parallel. This issue was also observed in SE(3)-equivariant networks, and was recently addressed by [21]. A promising future research direction is to develop similar acceleration techniques for BITR. Second, since BITR is deterministic, *i.e.*, it only predicts one result for a given input, it cannot handle symmetric PCs. Although this feature does not cause any difficulty in this work (there is no strictly symmetric PCs in this work due to noise, random sampling, etc), it may be problematic in future applications such as molecule modelling where symmetric PCs exist, *e.g.*, benzene rings. To address this issue, we plan to generalize BITR to a generative model in the future. More discussions can be found in Appx. E.

## Acknowledgments

This work is funded by the Swedish Research Council through grant agreement no. 2019-03686 and Chalmers AI Research Initiative (CHAIR). The computations were enabled by resources provided by the National Academic Infrastructure for Supercomputing in Sweden (NAISS), partially funded by the Swedish Research Council through grant agreement no. 2022-06725. The authors would like to thank Jan Gerken at Chalmers and Nan Xue at Ant Group for useful discussions, and thank the anonymous reviewers for their insightful comments and suggestions.

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

## A  SE(3)-equivariant Transformers

A well-known $SE(3)$-equivariant network is $SE(3)$-transformer [11], which adapts the powerful transformer structure [35] to $SO(3)$-equivariant settings. In this model, the feature map $f$ of each layer is defined as a tensor field supported on a 3-D PC:

$$f(x) = \sum_{u=1}^{M} f_u \delta(x - x_u), \tag{15}$$

where $\delta$ is the Dirac function, $X = \{x_u\}_{u=1}^{M} \subseteq \mathbb{R}^3$ is a point set, and $f_u$ is the feature attached to $x_u$. Here, feature $f_u$ takes the form of $f_u = \oplus_p f_u^p$, where the component $f_u^p \in V_p$ is the degree-$p$ feature, *i.e.*, it transforms according to $\rho_p$ under the action of $SO(3)$. For example, when $f_u$ represents the norm vector of a point cloud, then $f_u = f_u^1 \in \mathbb{R}^3$. We also write the collection of all degree-$p$ features at $x_u$ as $F^p(x_u) \in \mathbb{R}^{c \times (2p+1)}$, where $c$ is the number of channels.

For each transformer layer, the degree-$k$ output feature at point $x_i$ is computed by performing message passing:

$$f_{\text{out}}^l(x_u) = \underbrace{W^l F_{\text{in}}^l(x_u)}_{\text{self-interaction}} + \sum_l \sum_{v \in \mathcal{N}(u) \backslash \{u\}} \underbrace{\alpha_{uv}}_{\text{attention}} \underbrace{\mathbf{V}_{uv}^{lk}}_{\text{message}}, \tag{16}$$

where $\mathcal{N}(u)$ represents the neighborhood of $u$, $W^l \in \mathbb{R}^{1 \times c}$ is the learnable weight for self-interaction, $c$ represents the number of channels, and

$$\alpha_{uv} = \frac{\exp\left(\mathbf{Q}_u^\top \mathbf{K}_{uv}\right)}{\sum_{v'} \exp\left(\mathbf{Q}_u^\top \mathbf{K}_{uv'}\right)} \tag{17}$$

is the attention from $v$ to $u$. Here, key $\mathbf{K}$, value $\mathbf{V}$ and query $\mathbf{Q}$ are

$$\mathbf{Q}_u = \bigoplus_l W_Q^l F_{\text{in}}^l(x_u), \ \mathbf{K}_{uv} = \bigoplus_l \sum_k \mathcal{W}_K^{lk}(x_v - x_u) f_{\text{in}}^k(x_v), \ \mathbf{V}_{uv}^{lk} = \mathcal{W}_V^{lk}(x_v - x_u) f_{\text{in}}^k(x_v)$$
$$\tag{18}$$

where $W_Q^l \in \mathbb{R}^{1 \times (2l+1)}$ is a learnable weight, and the kernel $\mathcal{W}^{lk}(x) \in \mathbb{R}^{(2l+1) \times (2k+1)}$ is defined as

$$vec(\mathcal{W}^{lk}(x)) = \sum_{J=|k-l|}^{k+l} \underbrace{\varphi_J^{lk}(\|x\|)}_{\text{radial component}} \underbrace{Q_J^{lk} Y_J(x/\|x\|)}_{\text{angular component}}, \tag{19}$$

where $vec(\cdot)$ is the vectorize function, the learnable radial component $\varphi_J^{lk} : \mathbb{R}_+ \to R$ is parametrized by a neural network, and the non-learnable angular component is determined by Clebsch-Gordan constant $Q$ and the spherical harmonic $Y_J : \mathbb{R}^3 \to \mathbb{R}^{2J+1}$.

## B  Derive of the Convolutional Kernel

To derive the kernel (8) for a $SE(3) \times SE(3)$-transformer layer, we consider the equivariant convolution as a simplified version of the $SE(3) \times SE(3)$-transformer layer:

$$(\mathcal{W} * f)^{\boldsymbol{o}}(z_u) := \sum_{\substack{\boldsymbol{i} \\ v \in KNN(u) \backslash \{u\}}} \mathcal{W}^{\boldsymbol{o},\boldsymbol{i}}(z_v - z_u) f_{\text{in}}^{\boldsymbol{i}}(z_v), \tag{20}$$

*i.e.*, we only consider the message $\mathbf{V}_{uv}^{\boldsymbol{o},\boldsymbol{i}}$ while fixing the self-interaction weight $W^{\boldsymbol{o}} = 0$ and attention $\alpha_{uv} = 1$ in (5). To ensure the $SE(3) \times SE(3)$-equivariance of convolution (20), *i.e.*,

$$\left((g_1 \times g_2)(\mathcal{W}^{\boldsymbol{o},\boldsymbol{i}} * f)\right)(z) = \left(\mathcal{W}^{\boldsymbol{o},\boldsymbol{i}} * ((g_1 \times g_2)f)\right)(z), \tag{21}$$

the kernel $\mathcal{W}$ must satisfy a constraint:

$$\rho_{\boldsymbol{i}}(r_{12}) \otimes \rho_{\boldsymbol{o}}(r_{12}) \mathcal{W}(z) = \mathcal{W}(r_{12}z), \tag{22}$$

where we abbreviate $r_1 \times r_2$ as $r_{12}$, abbreviate $vec(\mathcal{W}^{\boldsymbol{o},\boldsymbol{i}}) \in \mathbb{R}^{(2i_1+1)(2i_2+1)(2o_1+1)(2o_2+1)}$ as $\mathcal{W}$, and assume $\|z^1\| = \|z^2\| = 1$ for simpler notations. Equation (22) is generally known as the kernel constraint, and its necessity and sufficiency can be proved in a verbatim way as Theorem 2 in [37],

so we omit the proof here. Now we can obtain the kernel (8) by solving this constraint. Note that a direct formulation is given in Theorem 2.1 in [5], but here we derive it in a less abstract way.

We first observe that $\mathcal{W}^*(z) = Y_{i_1}(z^1) \otimes Y_{i_2}(z^2) \otimes Y_{o_1}(z^1) \otimes Y_{o_2}(z^2)$ is a special solution to equation (22), where $Y_J(\cdot) \in \mathbb{R}^{2J+1}$ is the column vector consisting of degree-$J$ harmonics with order $m = -J, ...0, ..., J$ as each row element, because

$$\rho_{\boldsymbol{i}}(r_{12}) \otimes \rho_{\boldsymbol{o}}(r_{12})\mathcal{W}^*(z)$$

$$= \Big(\rho_{i_1}(r_1) \otimes \rho_{i_2}(r_2) \otimes \rho_{o_1}(r_1) \otimes \rho_{o_2}(r_2)\Big)\Big(Y_{i_1}(z^1) \otimes Y_{i_2}(z^2) \otimes Y_{o_1}(z^1) \otimes Y_{o_2}(z^2)\Big)$$

$$= \Big(\rho_{i_1}(r_1)Y_{i_1}(z^1)\Big) \otimes \Big(\rho_{i_2}(r_2)Y_{i_2}(z^2)\Big) \otimes \Big(\rho_{o_2}(r_1)Y_{o_2}(z^1)\Big) \otimes \Big(\rho_{o_2}(r_2)Y_{o_2}(z^2)\Big)$$

$$= Y_{i_1}(r_1 z^1) \otimes Y_{i_2}(r_2 z^2) \otimes Y_{o_1}(r_1 z^1) \otimes Y_{o_2}(r_2 z^2)$$

$$= \mathcal{W}^*(r_{12}z).$$

Then we point out that $\{Y_{J_1}^{m_1}(z^1)Y_{J_2}^{m_2}(z^2)\}_{m_1, m_2, J_2, J_1}$ is an orthogonal and complete basis for two variable square-integrable functions $L^2(X, Y) : S^2 \times S^2 \to \mathbb{R}$, where $S^2$ is the 2-D sphere. Thus, we can write each component of $\mathcal{W}^*(z)$ as a linear combination of this basis. Specifically, let $\mathcal{V}(z) = \oplus_{J_1, J_2} Y_{J_1}(z^1) \otimes Y_{J_2}(z^2)$, and let $(m_1, m_2, m_3, m_4)$ be the index of the element $Y_{i_1}^{m_1}(z^1)Y_{i_2}^{m_2}(z^2)Y_{o_1}^{m_3}(z^1)Y_{o_2}^{m_4}(z^2)$ in $\mathcal{W}^*$, and $(J_1, m_5, J_2, m_6)$ be the index of the element $Y_{J_1}^{m_5}(z^1)Y_{J_2}^{m_6}(z^2)$ in $\mathcal{V}$. We have $\mathcal{W}^* = P\mathcal{V}$, where

$$P_{m_1, m_2, m_3, m_4, J_1, m_5, J_2, m_6}$$

$$= \int Y_{i_1}^{m_1}(z^1)Y_{i_2}^{m_2}(z^2)Y_{o_1}^{m_3}(z^1)Y_{o_2}^{m_4}(z^2)Y_{J_1}^{m_5}(z^1)Y_{J_2}^{m_6}(z^2)dz^2 dz^1$$

$$= \Big(\int Y_{i_1}^{m_1}(z^1)Y_{o_1}^{m_3}(z^1)Y_{J_1}^{m_5}(z^1)dz^1\Big)\Big(\int Y_{i_2}^{m_2}(z^2)Y_{o_2}^{m_4}(z^2)Y_{J_2}^{m_6}(z^2)dz^2\Big)$$

$$= \langle i_1, m_1, o_1, m_3 | J_1, m_5\rangle\langle i_2, m_2, o_2, m_4 | J_2, m_6\rangle\boldsymbol{c}(J_1)\boldsymbol{c}(J_2). \tag{23}$$

Here, $\boldsymbol{c}(J_1)$ and $\boldsymbol{c}(J_2)$ are coefficients related to $J_1$ and $J_2$ respectively. $\langle i, m_1, o, m_3 | J, m_5\rangle$ is known as the Clebsch-Gordan coefficient, the product $\langle i_1, m_1, o_1, m_3 | J_1, m_5\rangle\langle i_2, m_2, o_2, m_4 | J_2, m_6\rangle$ is known as the 2-nd order Clebsch-Gordan coefficient, and we represent it as $\mathcal{Q}$. In other words, we have $P = \mathcal{Q}\boldsymbol{c}$, where $\boldsymbol{c}$ is a block diagonal matrix, and each block is $\boldsymbol{c}(J_1)\boldsymbol{c}(J_2)I$. Therefore, we have

$$\Big(\rho_{\boldsymbol{i}}(r_{12}) \otimes \rho_{\boldsymbol{o}}(r_{12})\Big)P\mathcal{V}(z) = P\mathcal{V}(r_{12}z)$$

$$\Big(\rho_{\boldsymbol{i}}(r_{12}) \otimes \rho_{\boldsymbol{o}}(r_{12})\Big)\mathcal{Q}\boldsymbol{c}\mathcal{V}(z) = \mathcal{Q}\boldsymbol{c}\mathcal{V}(r_{12}z)$$

$$\mathcal{Q}^T\Big(\rho_{\boldsymbol{i}}(r_{12}) \otimes \rho_{\boldsymbol{o}}(r_{12})\Big)\mathcal{Q}\boldsymbol{c}\mathcal{V}(z) = \boldsymbol{c}\mathcal{V}(r_{12}z).$$

Since

$$\Big[\bigoplus_{J_1, J_2} \rho_{J_1, J_2}(r_{12})\Big]\boldsymbol{c}\mathcal{V}(z) = \boldsymbol{c}\mathcal{V}(r_{12}z) \tag{24}$$

for arbitrary $z$, we further have

$$\mathcal{Q}^T\Big(\rho_{\boldsymbol{i}}(r_{12}) \otimes \rho_{\boldsymbol{o}}(r_{12})\Big)\mathcal{Q} = \Big[\bigoplus_{J_1, J_2} \rho_{J_1, J_2}(r_{12})\Big] \tag{25}$$

by equating the above two equations. Finally, we stack this decomposition back to the kernel constraint (22), and obtain

$$\Big[\bigoplus_{J_1, J_2} \rho_{J_1, J_2}(r_{12})\Big]\mathcal{Q}^T\mathcal{W}(z) = \mathcal{Q}^T\mathcal{W}(r_{12}z), \tag{26}$$

which suggests that

$$\mathcal{Q}^T\mathcal{W}(z) = \boldsymbol{c}'\mathcal{V}(x), \tag{27}$$

$$\mathcal{W}(z) = \mathcal{Q}\boldsymbol{c}'\mathcal{V}(x), \tag{28}$$

where $\boldsymbol{c}'$ is the coefficient matrix of the same shape as $\boldsymbol{c}$, and each coefficient is arbitrary. Specifically,

$$\mathcal{W}(z) = \boldsymbol{c}'_{J_1, J_2}\sum_{J_1, J_2} \mathcal{Q}_{J_1, J_2}Y_{J_1}(z^1) \otimes Y_{J_2}(z^2). \tag{29}$$

We note that $\boldsymbol{c}'_{J_1, J_2}$ is parametrized by a neural network in our model (8).

## C  Proofs and theoretical results

### C.1  The proof of the equivariance of Arun's method

We first establish the result on the uniqueness of Arun's method. We begin with the result of the uniqueness of singular vectors.

**Lemma C.1.** *Let $A = U\Sigma V^T \in \mathbb{R}^{3\times 3}$ be the SVD decomposition. If a singular value $\sigma_j$ is distinct, then the corresponding singular vectors $U_j$ and $V_j$ can be determined up to a sign. If $\sigma_j \neq 0$, $U_j V_j^T$ is unique.*

*Proof.* If $\sigma_j \neq 0$, we have

$$A^T A = V\Sigma^2 V^T = V\Sigma^2 V^{-1}, \tag{30}$$

*i.e.*, the eigenvalues of $A^T A$ are $\sigma_i^2$, $i = 1, 2, 3$, and $V_i$ are the corresponding eigenvectors. Since $\sigma_j$ is distinct and all $\sigma_i \geq 0$, $\sigma_j^2$ is distinct. As a result, the eigenspace corresponding to $\sigma_j^2$ has dim 1, *i.e.*, it is spanned by $V_j$. Since $V_j$ is a unit vector, it can be determined up to a sign. In addition, we have $AV_j = \sigma_j U_j$, thus $U_j$ can also be determined up to a sign. $U_j V_j^T$ is unique since flipping the sign of $V_j$ always leads to the flipping of sign of $U_j$, and vice versa.

If $\sigma_j = 0$, we can still repeat the above argument to determine $V_j$ up to a sign, and determine $U_j$ up to a sign using the above argument for $AA^T$. $\qquad\square$

Note that in Arun's algorithm, we use the SVD projection defined as follows.

$$\overline{SVD}(A) = \hat{U} diag(1, 1, sign(det(\hat{U}\hat{V}^T)))\hat{V}^T, \tag{31}$$

where $A = \hat{U}\hat{\Sigma}\hat{V}^T$, $\hat{U}, \hat{V} \in O(3)$ is the SVD decomposition of $A$. In other words, if $det(\hat{U}\hat{V}^T) = 1$, then we take $\hat{U}\hat{V}^T$, otherwise we flip the sign of $\hat{U}_3\hat{V}_3$. An important observation is that this SVD projection is unique under a mild assumption.

**Assumption C.2.** In the SVD decomposition, $\sigma_3$ is distinct, *i.e.*, $\sigma_1 \geq \sigma_2 > \sigma_3 \geq 0$.

**Proposition C.3** (Uniqueness of SVD projection). *Let $A = U\Sigma V^T \in \mathbb{R}^{3\times 3}$ be the SVD decomposition, where $\Sigma = diag(\sigma_1, \sigma_2, \sigma_3)$ is a diagonal matrix, and $U, V \in O(3)$. If assumption C.2 is satisfied, then $\overline{SVD}(A)$ is unique.*

*Proof.* Let $A = \tilde{U}\tilde{\Sigma}\tilde{V}^T$ be another SVD decomposition of $A$, *i.e.*,

$$A = \sigma_1 U_1 V_1^T + \sigma_2 U_2 V_2^T + \sigma_3 U_3 V_3^T = \sigma_1 \tilde{U}_1 \tilde{V}_1^T + \sigma_2 \tilde{U}_2 \tilde{V}_2^T + \sigma_3 \tilde{U}_3 \tilde{V}_3^T, \tag{32}$$

the goal is to prove

$$U_1 V_1^T + U_2 V_2^T + sign(det(UV^T))U_3 V_3^T = \tilde{U}_1 \tilde{V}_1^T + \tilde{U}_2 \tilde{V}_2^T + sign(det(\tilde{U}\tilde{V}^T))\tilde{U}_3 \tilde{V}_3^T. \tag{33}$$

We first show that $U_1 V_1^T + U_2 V_2^T = \tilde{U}_1 \tilde{V}_1^T + \tilde{U}_2 \tilde{V}_2^T$.

1) If $\sigma_3 = 0$ and $\sigma_1 = \sigma_2$, then

$$U_1 V_1^T + U_2 V_2^T = \tilde{U}_1 \tilde{V}_1^T + \tilde{U}_2 \tilde{V}_2^T = \frac{1}{\sigma_1} A. \tag{34}$$

2) If $\sigma_3 = 0$ and $\sigma_1 > \sigma_2$, then $\sigma_1$ and $\sigma_2$ are distinct and nonzero. According to lemma C.1, $U_1 V_1^T = \tilde{U}_1 \tilde{V}_1^T$ and $U_2 V_2^T = \tilde{U}_2 \tilde{V}_2^T$, thus the summation of these two terms is equal.

3) If $\sigma_3 > 0$ and $\sigma_1 > \sigma_2$, the argument is the same as 2)

4) If $\sigma_3 > 0$ and $\sigma_1 = \sigma_2$, then $\sigma_3$ is distinct and nonzero. According to lemma C.1, $U_3 V_3^T = \tilde{U}_3 \tilde{V}_3^T$. Thus,

$$U_1 V_1^T + U_2 V_2^T = \tilde{U}_1 \tilde{V}_1^T + \tilde{U}_2 \tilde{V}_2^T = \frac{1}{\sigma_1}(1 - \sigma_3 U_3 V_3^T). \tag{35}$$

Thus we have $U_1 V_1^T + U_2 V_2^T = \tilde{U}_1 \tilde{V}_1^T + \tilde{U}_2 \tilde{V}_2^T$.

Now we prove that the last term in Eqn. (33) is equal. We have $\tilde{U}_3\tilde{V}_3^T = \pm U_3 V_3^T$, and we have shown that $U_1 V_1^T + U_2 V_2^T = \tilde{U}_1\tilde{V}_1^T + \tilde{U}_2\tilde{V}_2^T$, thus we have

$$
\begin{aligned}
det(\tilde{U}_1\tilde{V}_1^T + \tilde{U}_2\tilde{V}_2^T + \tilde{U}_3\tilde{V}_3^T) &= det(U_1 V_1^T + U_2 V_2^T \pm U_3 V_3^T) \\
&= det(U)det([V_1, V_2, \pm V_3]^T) \\
&= det(U)\Big(\pm det([V_1, V_2, V_3]^T)\Big) \\
&= \pm det(U_1 V_1^T + U_2 V_2^T + U_3 V_3^T)
\end{aligned}
\tag{36}
$$

By multiplying this term with $\tilde{U}_3\tilde{V}_3^T = \pm U_3 V_3^T$, we have

$$
det(\tilde{U}_1\tilde{V}_1^T + \tilde{U}_2\tilde{V}_2^T + \tilde{U}_3\tilde{V}_3^T)\tilde{U}_3\tilde{V}_3^T = det(U_1 V_1^T + U_2 V_2^T + U_3 V_3^T)U_3 V_3^T,
\tag{37}
$$

which suggests that last term in Eqn. (33) is equal. In summary, we have proved (33). $\square$

Finally, we can prove the uniqueness of Arun's method under the same assumption.

**Proposition C.4** (Uniqueness of Arun's method). *Under assumption C.2, Arun's method (1) is unique.*

*Proof.* According to Prop. C.3, $r = SVD(\bar{r})$ is unique. As a result, $t = \boldsymbol{m}(Y) - r\boldsymbol{m}(X)$ is also unique. $\square$

Throughout this work, we assume that assumption C.2 holds, so that the uniqueness of Arun's method and BITR can be guaranteed.

For the rest of this appendix, we use a simpler notation of SVD projection, where we simply absorb the sign matrix into $\hat{U}$ and $\hat{V}$:

$$
U = \big(\hat{U}diag(1, 1, sign(det(\hat{U})))\big) \in SO(3),
\tag{38}
$$

$$
V^T = \big(diag(1, 1, sign(det(\hat{V})))\hat{V}^T\big) \in SO(3),
\tag{39}
$$

$$
and \quad \Sigma = diag(\hat{\sigma}_1, \hat{\sigma}_2, \hat{\sigma}_3 sign(det(\hat{U}\hat{V}^T))),
\tag{40}
$$

thus obtain the following equivalent definition.

**Definition C.5.** The SVD projection of matrix $A$ is defined as

$$
SVD(A) = UV^T
\tag{41}
$$

where $A = U\Sigma V^T$ and $U, V \in SO(3)$.

Finally, we can discuss the equivariance of Arun's method.

**Lemma C.6.** *Let*

$$
SVD(A) = UV^T
\tag{42}
$$

*be the SVD projection of A, i.e., A can be decomposed as $A = U\Sigma V^T$, $\Sigma$ is a diagonal matrix and $U, V \in SO(3)$. We have*

$$
SVD(r_2 A r_1^T) = r_2 SVD(A) r_1^T,
\tag{43}
$$

$$
SVD(A^T) = (SVD(A))^T,
\tag{44}
$$

$$
SVD(cA) = SVD(A),
\tag{45}
$$

*for arbitrary $A \in GL(3)$, $r_1, r_2 \in SO(3)$, and $c \in \mathbb{R}_+$.*

*The Proof of Lemma. C.6.* We have

$$
r_2 A r_1^T = r_2 U\Sigma V^T r_1^T = (r_2 U)\Sigma(r_1 V)^T,
\tag{46}
$$

where $r_2 U \in SO(3)$, $r_1 V \in SO(3)$ and $\Sigma$ is a diagonal matrix. In other words, (46) is a *SVD* decomposition of matrix $r_2 A r_1^T$, thus the *SVD* projection can be computed as

$$
SVD(r_2 A r_1^T) = (r_2 U)(r_1 V)^T = r_2(UV^T)r_1^T = r_2 SVD(A)r_1^T,
\tag{47}
$$

which proves the first part of this lemma. We omit the other two statements of this lemma since they can be proved similarly. $\square$

*The Proof of Prop. 3.2.* Let $\Phi_A$ be Arun's method and $\Phi_A(X, Y) = (r(X, Y), t(X, Y)) = g$. We directly verify these three equivariances according to Def. 3.1. We only prove the $SE(3)$-bi-equivariance and swap-equivariance and omit the proof of scale-equivariance, because it can be proved similarity.

**1) $SE(3)$-bi-equivariance**  We compute $\Phi_A(g_1 X, g_2 Y)$ for the perturbed PCs $g_1 X = \{r_1 x_i + t_1\}_{i=1}^N$ and $g_2 Y = \{r_2 y_i + t_2\}_{i=1}^N$ as follows. We first compute $\bar{r}(g_1 X, g_2 Y)$ as

$$\bar{r}(g_1 X, g_2 Y) = \sum_i (r_2 \bar{y}_i)(\bar{x}_i^T r_1^T) = r_2 \left( \sum_i \bar{y}_i \bar{x}_i^T \right) r_1^T = r_2 \bar{r}(X, Y) r_1^T. \tag{48}$$

Then we compute $r$ via *SVD*:

$$r(g_1 X, g_2 Y) = SVD(\bar{r}(g_1 X, g_2 Y)) = SVD(r_2 \bar{r}(X, Y) r_1^T) = r_2 SVD(\bar{r}(X, Y)) r_1^T = r_2 r(X, Y) r_1^T, \tag{49}$$

where the 3-rd equality holds due to Lemma. C.6. Since the mean values are also perturbed as $m(g_1 X) = g_1 m(X)$ and $m(g_2 Y) = g_2 m(Y)$, we compute $t(g_1 X, g_2 Y)$ as

$$\begin{aligned} t(g_1 X, g_2 Y) = m(g_2 Y) - r(g_1 X, g_2 Y) m(g_1 X) &= g_2 m(Y) - r_2 r(X, Y) r_1^T g_1 m(X) \\ &= r_2 t(X, Y) - r_2 r(X, Y) r_1^{-1} t(X, Y) + t_2. \end{aligned} \tag{50}$$

In summary,

$$\Phi_A(g_1 X, g_2 Y) = (r_2 r(X, Y) r_1^{-1}, -r_2 r(X, Y) r_1^{-1} t_1 + r_2 t(X, Y) + t_2) = g_2 g g_1^{-1}, \tag{51}$$

which proves the $SE(3)$-bi-equivariance of Arun's method.

**2) swap-equivariance**  We compute $\Phi_A(Y, X)$ as follows. We first compute $\bar{r}(Y, X)$ as

$$\bar{r}(Y, X) = \sum_i \bar{x}_i \bar{y}_i^T = \left( \sum_i \bar{y}_i^T \bar{x}_i \right)^T = (\bar{r}(X, Y))^T. \tag{52}$$

Then we compute $r(Y, X)$ via *SVD*:

$$r(Y, X) = SVD(\bar{r}(Y, X)) = SVD\left( (\bar{r}(X, Y))^T \right) = (SVD(\bar{r}(X, Y)))^T = r(X, Y)^T, \tag{53}$$

We can finally compute $t(Y, X)$ as

$$t(Y, X) = m(X) - r(Y, X) m(Y) = m(X) - r(X, Y)^{-1} m(Y) = -r(X, Y)^{-1} t(X, Y). \tag{54}$$

In summary,

$$\Phi_A(Y, X) = (r(X, Y)^{-1}, -r(X, Y)^{-1} t(X, Y)) = g^{-1}, \tag{55}$$

which proves the swap-equivariance of Arun's method.

$\square$

### C.2   The proof of the equivariance of BITR

Before proving the $SE(3)$-bi-equivariance of BITR, we first verify that each layer in a $SE(3) \times SE(3)$-transformer is indeed $SE(3) \times SE(3)$-equivariant. Intuitively, the equivariance of a transformer layer (5) is a result of the kernel design, *i.e.*, the kernel constraint (22), and the invariance of the attention. We state this result in the following lemma for completeness, and we note that similar techniques are also used in the construction of $SE(3)$-transformer [11].

**Lemma C.7.** *The transformer layer (5) is $SE(3) \times SE(3)$-equivariant.*

*Proof.* We abbreviate $r_1 \times r_2$ as $r_{12}$, and $g_1 \times g_2$ as $g_{12}$. Let $L$ be a transformer layer (5), we seek to prove $(g_{12})(L(f)) = L((g_{12})(f))$ for the input tensor field $f$ and $g_1, g_2 \in SE(3)$. We compute the

RHS of the equation as

$$\left(L(g_{12}(f))\right)^{\boldsymbol{o}}(z_u)$$

$$=W^{\boldsymbol{o}}\big(g_{12}(F)\big)^{\boldsymbol{o}}(z_u) + \sum_{\substack{i \\ z_v \in supp(g_{12}(f))}} \alpha(g_{12}(f), z_u, z_v)\mathcal{W}^{\boldsymbol{o},i}(z_v - z_u)\big(g_{12}(f)\big)^i(z_v)$$

$$=W^{\boldsymbol{o}}F^{\boldsymbol{o}}(g_{12}^{-1}z_u)\big(\rho_{\boldsymbol{o}}(r_{12})\big)^T + \sum_{\substack{i \\ z_v \in supp(g_{12}(f))}} \alpha(g_{12}(f), z_u, z_v)\mathcal{W}^{\boldsymbol{o},i}(z_v - z_u)\rho_i(r_{12})f^i(g_{12}^{-1}z_v)$$

$$=\underbrace{W^{\boldsymbol{o}}F^{\boldsymbol{o}}(g_{12}^{-1}z_u)\big(\rho_{\boldsymbol{o}}(r_{12})\big)^T}_{\textcircled{A}} + \sum_{\substack{i \\ z_v \in supp(f)}} \underbrace{\alpha(g_{12}(f), z_u, g_{12}z_v)}_{} \underbrace{\mathcal{W}^{\boldsymbol{o},i}(g_{12}z_v - z_u)}_{\textcircled{B}} \underbrace{\rho_i(r_{12})f^i(z_v)}_{\textcircled{C}}.$$

Note that here we have $g_{12}F^{\boldsymbol{o}}(z) = F^{\boldsymbol{o}}(g_{12}^{-1}z)(\rho_{\boldsymbol{o}}(r_{12}))^T$ because we write $F$ in the channel-first form, $i.e.$, the shape of $F^{\boldsymbol{o}}$ is $c \times (2o_1 + 1)(2o_2 + 1)$. The LHS of the equation is computed as

$$\left(g_{12}\big(L(f)\big)\right)^{\boldsymbol{o}}(z_u)$$

$$=\underbrace{W^{\boldsymbol{o}}F^{\boldsymbol{o}}(g_{12}^{-1}z_u)\big(\rho_{\boldsymbol{o}}(r_{12})\big)^T}_{\textcircled{A'}} + \sum_{\substack{i \\ z_v \in supp(f)}} \underbrace{\alpha(f, g_{12}^{-1}z_u, z_v)}_{\textcircled{B'}} \underbrace{\rho_{\boldsymbol{o}}(r_{12})\mathcal{W}^{\boldsymbol{o},i}(z_v - g_{12}^{-1}z_u)f^i(z_v)}_{\textcircled{C'}}.$$

We now verify that these 3 terms are equal respectively.

$\textcircled{C} = \textcircled{C'}$: By design, the kernel $\mathcal{W}$ satisfies the kernel constraint (22)

$$\rho_{\boldsymbol{o}}(r_{12})\mathcal{W}^{\boldsymbol{o},i}(z)\rho_i^{-1}(r_{12}) = \mathcal{W}(r_{12}z). \tag{56}$$

In addition, let $z_u' = g_{12}^{-1}z_u$. We have

$$g_{12}z_v - z_u = g_{12}z_v - g_{12}z_u' = r_{12}(z_v - z_u') = r_{12}(z_v - g_{12}^{-1}z_u). \tag{57}$$

Thus, we can obtain the equality by combining the above two equations.

$\textcircled{B} = \textcircled{B'}$: We compute $\mathbf{Q}$ and $\mathbf{K}$ for $\textcircled{B}$ and $\textcircled{B'}$ respectively. We have

$$\mathbf{Q}(g_{12}(f), z_u, g_{12}z_v) = \bigoplus_{\boldsymbol{o}} W_Q^{\boldsymbol{o}}(g_{12}F)^{\boldsymbol{o}}(z_u) = \bigoplus_{\boldsymbol{o}} W_Q^{\boldsymbol{o}}F^{\boldsymbol{o}}(r_{12}^{-1}(z_u))\big(\rho_{\boldsymbol{o}}(r_{12})\big)^T,$$

$$\mathbf{Q}(f, g_{12}^{-1}z_u, z_v) = \bigoplus_{\boldsymbol{o}} W_Q^{\boldsymbol{o}}F^{\boldsymbol{o}}(g_{12}^{-1}(z_u)),$$

$$\mathbf{K}(g_{12}(f), z_u, g_{12}z_v) = \bigoplus_{\boldsymbol{o}} \sum_i \mathcal{W}_K^{\boldsymbol{o},i}(g_{12}z_v - z_u)\big(g_{12}(f)\big)^i(g_{12}z_v)$$

$$= \bigoplus_{\boldsymbol{o}} \sum_i \mathcal{W}_K^{\boldsymbol{o},i}(g_{12}z_v - z_u)\rho_i(r_{12})f^i(z_v),$$

and

$$\mathbf{K}(f, g_{12}^{-1}z_u, z_v) = \bigoplus_{\boldsymbol{o}} \sum_i \mathcal{W}_K^{\boldsymbol{o},i}(z_v - g_{12}^{-1}z_u)f^i(z_v)$$

$$= \bigoplus_{\boldsymbol{o}} \sum_i \big(\rho_{\boldsymbol{o}}(r_{12})\big)^T \mathcal{W}_K^{\boldsymbol{o},i}(g_{12}z_v - z_u)\rho_i(r_{12})f^i(z_v),$$

where the last equation holds due to the kernel constraint (22) and the orthogonality of $\rho_{\boldsymbol{o}}(r_{12})$. Specifically, $\rho_{\boldsymbol{o}}(r_{12}) = \rho_{o_1}(r_1) \otimes \rho_{o_2}(r_2)$ is an orthogonal matrix because $\rho_{o_1}$ and $\rho_{o_2}$ are orthogonal representations, $i.e.$, $\rho_{o_1}(r_1)$ and $\rho_{o_2}(r_2)$ are orthogonal matrices.

Thus, we have

$$\langle \mathbf{Q}(g_{12}(f), z_u, g_{12}z_v), \mathbf{K}(g_{12}(f), z_u, g_{12}z_v)\rangle$$

$$= \sum_{\boldsymbol{o}} W_Q^{\boldsymbol{o}} F^{\boldsymbol{o}}(r_{12}^{-1}(z_u))\Big(\rho_{\boldsymbol{o}}(r_{12})\Big)^T \Big(\sum_{\boldsymbol{i}} \mathcal{W}_K^{\boldsymbol{o},\boldsymbol{i}}(g_{12}z_v - z_u)\rho_{\boldsymbol{i}}(r_{12})f^{\boldsymbol{i}}(z_v)\Big)$$

$$= \sum_{\boldsymbol{o}} W_Q^{\boldsymbol{o}} F^{\boldsymbol{o}}(r_{12}^{-1}(z_u))\Big(\sum_{\boldsymbol{i}} \Big(\rho_{\boldsymbol{o}}(r_{12})\Big)^T \mathcal{W}_K^{\boldsymbol{o},\boldsymbol{i}}(g_{12}z_v - z_u)\rho_{\boldsymbol{i}}(r_{12})f^{\boldsymbol{i}}(z_v)\Big)$$

$$= \langle \mathbf{Q}(f, g_{12}^{-1}z_u, z_v), \mathbf{K}(f, g_{12}^{-1}z_u, z_v)\rangle$$

Finally, $\widehat{A} = \widehat{A'}$ is trivial. In summary, we have shown $(g_{12})(L(f)) = L((g_{12})(f))$, which proves the $SE(3) \times SE(3)$-equivariance of the transformer layer (5). $\qquad\square$

On the other hand, an Elu layer (5) is also $SE(3) \times SE(3)$-equivariant, because this layer is piecewise linear. We state this statement formally in the following lemma, and omit the proof. We note that a similar argument was used in [9].

**Lemma C.8.** *The Elu layer (9) is $SE(3) \times SE(3)$-equivariant.*

Now we can prove the $SE(3)$-bi-equivariance of BITR.

*The Proof of Prop. 4.1.* Let $g = \Phi_P \circ \Phi_S(X, Y)$, and $f_{out} = \Phi_S(X, Y)$. We prove this proposition by showing

$$g_2^{-1}gg_1 = \Phi_P \circ \Phi_S(g_1X, g_2Y). \tag{58}$$

We compute the RHS of the equation as follows. First, since each point in $\tilde{X}$ and $\tilde{Y}$ is a convex combination of $X$ and $Y$ respectively, we have

$$\tilde{X}(g_1X, g_2Y) = g_1\tilde{X}(X, Y) \quad \tilde{Y}(g_1X, g_2Y) = g_2\tilde{Y}(X, Y). \tag{59}$$

Then we have $Z(g_1X, g_2Y) = (g_1 \times g_2)Z(X, Y)$ because $Z = \tilde{X} \oplus \tilde{Y}$. When $Z(g_1X, g_2Y)$ is fed to the $SE(3) \times SE(3)$-transformer, the output feature will be $(g_1 \times g_2)f_{out}$ by design (This is verified in Lemma. C.7 and Lemma. C.8). In particular, for degree-$(1, 1)$ feature $\tilde{r}$, degree-$(1, 0)$ feature $t_X$ and degree-$(0, 1)$ feature $t_Y$, we have

$$\tilde{r}(g_1X, g_2Y) = (r_1 \otimes r_2)\tilde{r}(X, Y) \quad t_X(g_1X, g_2Y) = r_1t_X(X, Y) \quad t_Y(g_1X, g_2Y) = r_2t_Y(X, Y). \tag{60}$$

Therefore, we have $\hat{r}(g_1X, g_2Y) = r_2\tilde{r}(X, Y)r_1^{-1}$ by applying *unvec*$(\cdot)$ to the first equation, *i.e.*, $\hat{r} = $ *unvec*$(\tilde{r})$. Finally, we compute projection $\Phi_P$ similar to proof C.1:

$$r(g_1X, g_2Y) = r_2r(X, Y)r_1^T, \tag{61}$$

and

$$\begin{aligned} t(g_1X, g_2Y) &= \boldsymbol{m}(g_2Y) + t_Y(g_1X, g_2Y) - r(g_1X, g_2Y)(\boldsymbol{m}(g_1X) + t_X(g_1X, g_2Y)) \\ &= g_2\boldsymbol{m}(Y) + r_2t_Y(X, Y) - r_2r(X, Y)r_1^T(g_1\boldsymbol{m}(X) + r_1t_X(X, Y)) \\ &= r_2t(X, Y) - r_2r(X, Y)r_1^{-1}t(X, Y) + t_2. \end{aligned}$$

In summary,

$$\Phi_P \circ \Phi_S(g_1X, g_2Y) = (r_2r(X, Y)r_1^{-1}, -r_2r(X, Y)r_1^{-1}t_1 + r_2t(X, Y) + t_2) = g_2gg_1^{-1}, \tag{62}$$

which proves the $SE(3)$-bi-equivariance of BITR. $\qquad\square$

We finally prove Prop. 5.1 and Prop. 5.3 similarly to Prop. 3.2.

*The Proof of Prop. 5.1.* We compute $\Phi_P(s(f))$ as follows. First, we have

$$(s(f))^{1,1}(z) = \big(f^{1,1}(s(z))\big)^T, \ (s(f))^{1,0}(z) = \big(f^{0,1}(s(z))\big)^T, \ (s(f))^{0,1}(z) = \big(f^{1,0}(s(z))\big)^T \tag{63}$$

by definition. Then we compute the equivariant features as

$$\hat{r}(s(f)) = (\hat{r}(f))^T, \quad t_X(s(f)) = t_Y(f), \quad t_Y(s(f)) = t_X(f), \tag{64}$$

where we treat $t_X$ and $t_Y$ as vectors. Finally, the output can be computed as

$$r(s(f)) = SVD(\hat{r}(s(f))) = SVD((\hat{r}(f))^T) = (SVD(\hat{r}(f)))^T = (r(f))^T \tag{65}$$

according to lemma C.6, and

$$\begin{aligned}
t(s(f)) &= \boldsymbol{m}(Y(s(f))) + t_Y(s(f)) - r(s(f))(\boldsymbol{m}(X(s(f))) + t_X(s(f))) \\
&= \boldsymbol{m}(X) + t_X(f) - (r(f))^T (\boldsymbol{m}(Y) + t_Y(f)) \\
&= -r(f)^T t(f).
\end{aligned}$$

In other words, $\Phi_P(s(f)) = (r^T, -r^T t) = g^{-1}$, which proves this proposition. $\qquad\square$

*The Proof of Prop. 5.3.* We compute $\Phi_P(c(f))$ as follows. First, since $f$ is a degree-1 $\mathbb{R}_+$-equivariant tensor field, $f$ satisfies $c(f)(z) = cf(c^{-1}z)$. Therefore, we have

$$\hat{r}(c(f)) = (c\hat{r}(f)), \quad t_X(c(f)) = ct_X(f), \quad t_Y(c(f)) = t_Y(f) \tag{66}$$

by definition. Then, we compute the output as

$$r(c(f)) = SVD(\hat{r}(c(f))) = SVD((c\hat{r}(f))) = (SVD(\hat{r}(f))) = (r(f)) \tag{67}$$

according to lemma C.6, and

$$\begin{aligned}
t(c(f)) &= \boldsymbol{m}(Y(c(f))) + t_Y(c(f)) - r(c(f))(\boldsymbol{m}(X(c(f))) + t_X(c(f))) \\
&= c\boldsymbol{m}(Y) + ct_Y(f) - r(f)(c\boldsymbol{m}(Y) + ct_Y(f)) \\
&= ct(f).
\end{aligned}$$

In other words, $\Phi_P(c(f)) = (r, ct)$, which proves this proposition. $\qquad\square$

## C.3 More results of Sec. 5.1

### C.3.1 The proof of Prop. 5.2

In this subsection, we use slightly different notations than other parts of the paper. We regard the kernel $\mathcal{W}^{o,i}$ as a 4-D tensor of shape $(2o_1 + 1) \times (2o_2 + 1) \times (2i_1 + 1) \times (2i_2 + 1)$, and we regard the feature $f^i$ as a 2-D tensor of shape $(2i_1 + 1) \times (2i_2 + 1)$, *i.e.*, a matrix, when it is multiplied by a kernel. Therefore, the multiplication $\mathcal{W}^{o,i} f^i$ is treated as a tensor product where all two dimensions of $f^i$ are treated as rows, *i.e.*, the result of $\mathcal{W}^{o,i} f^i$ is of shape $(2o_1 + 1) \times (2o_2 + 1)$. Similarly, we regard the collection of features $F^o$ as a 3-D tensor of shape $c \times (2o_1 + 1) \times (2o_2 + 1)$. When it is multiplied by a self-interaction weight $W^o \in \mathbb{R}^{1 \times c}$, the result is $W^o F^o \in \mathbb{R}^{(2o_1+1) \times (2o_2+1)}$.

We first make the following observation on the symmetry of the kernel.

**Lemma C.9.** *If the radial function $\varphi$ satisfies $\varphi_{J_1,J_2}^{i,o}(\|z^1\|, \|z^2\|) = \varphi_{J_2,J_1}^{\tilde{i},\tilde{o}}(\|z^2\|, \|z^1\|)$ for all $z^1$, $z^2$, $J_1$, $J_2$, then kernel $\mathcal{W}$ (5) satisfies*

$$\mathcal{W}^{o,i}(z) = \left(\mathcal{W}^{\tilde{o},\tilde{i}}(s(z))\right)^{T_{12}T_{34}}, \tag{68}$$

*where $T_{ij}$ represents the transpose of the $i$-th and $j$-th dimension of a tensor.*

*Proof.* According to (5) and the discussion in Sec. B, the $(m_3, m_4, m_1, m_2)$-th element of $\mathcal{W}^{o,i}(z)$ is

$$\sum_{J_1=|o_1-i_1|}^{o_1+i_1} \sum_{J_2=|o_2-i_2|}^{o_2+i_2} \sum_{m_5=-J_1}^{J_1} \sum_{m_6=-J_2}^{J_2} \varphi_{J_1,J_2}^{i,o}(\|z^1\|, \|z^2\|)\langle i_1, m_1, o_1, m_3|J_1, m_5\rangle$$
$$\langle i_2, m_2, o_2, m_4|J_2, m_6\rangle Y_{J_1}^{m_5}(z^1/\|z^1\|) Y_{J_2}^{m_6}(z^2/\|z^2\|), \tag{69}$$

and the $(m_4, m_3, m_2, m_1)$-th element of $\mathcal{W}^{\tilde{o},\tilde{i}}(s(z))$ is

$$\sum_{J_2=|o_2-i_2|}^{o_2+i_2} \sum_{J_1=|o_1-i_1|}^{o_1+i_1} \sum_{m_6=-J_2}^{J_2} \sum_{m_5=-J_1}^{J_1} \varphi_{J_2,J_1}^{\tilde{i},\tilde{o}}(\|z^2\|, \|z^1\|)\langle i_2, m_2, o_2, m_4|J_2, m_6\rangle$$

$$\langle i_1, m_1, o_1, m_3|J_1, m_5\rangle Y_{J_2}^{m_6}(z^2/\|z^2\|)Y_{J_1}^{m_5}(z^1/\|z^1\|). \quad (70)$$

Since the angular components in (69) and (70) are the same, and the radial components are equal by assumption, we immediately conclude that (69) and (70) are equal. In other words,

$$\left(\mathcal{W}^{o,i}(z)\right)_{m_3,m_4,m_1,m_2} = \mathcal{W}^{\tilde{o},\tilde{i}}(s(z))_{m_4,m_3,m_2,m_1}, \quad (71)$$

which proves this lemma. $\qquad\square$

Then we proceed to the proof of Prop. 5.2.

*The Proof of Prop. 5.2.* Let $L$ be a transformer layer (5), we seek to prove $s(L(f)) = L(s(f))$ for the input tensor field $f$.

To this end, we expand the RHS of the equation as

$$\left(L(s(f))\right)^o(z_u)$$

$$= W^o\left(s(F)\right)^o(z_u) + \sum_{\substack{i\in I(s(f))\\z_v\in supp(s(f))}} \alpha(s(f), z_u, z_v)\mathcal{W}^{o,i}(z_v-z_u)\left(s(f)\right)^i(z_v)$$

$$= W^o\left(F^{\tilde{o}}(s(z_u))\right)^{T_{23}} + \sum_{\substack{i\in I(s(f))\\z_v\in supp(s(f))}} \alpha(s(f), z_u, z_v)\mathcal{W}^{o,i}(z_v-z_u)\left(f^{\tilde{i}}(s(z_v))\right)^T$$

$$= \underbrace{W^o\left(F^{\tilde{o}}(s(z_u))\right)^{T_{23}}}_{\textcircled{A}} + \sum_{\substack{i\in I(f)\\z_v\in supp(f)}} \underbrace{\alpha(s(f), z_u, s(z_v))}_{\textcircled{B}}\underbrace{\mathcal{W}^{o,\tilde{i}}(s(z_v)-z_u)\left(f^i(z_v)\right)^T}_{\textcircled{C}},$$

where $I(f)$ is the set of all degrees of field $f$, and $supp(f)$ is the support of field $f$. The last equation holds because we replace $z_v$ by $s(z_v)$, and replace $i$ by $\tilde{i}$. We expand the LHS of the equation as

$$\left(s(L(f))\right)^o(z_u) = \left(\left(L(f)\right)^{\tilde{o}}(s(z_u))\right)^T$$

$$= \underbrace{\left(W^{\tilde{o}}F^{\tilde{o}}(s(z_u))\right)^T}_{\textcircled{A'}} + \sum_{\substack{i\in I(f)\\z_v\in supp(f)}} \underbrace{\alpha(f, s(z_u), z_v)}_{\textcircled{B'}}\underbrace{\left(\mathcal{W}^{\tilde{o},i}(z_v-s(z_u))f^i(z_v)\right)^T}_{\textcircled{C'}}.$$

We now verify that these three terms are equal respectively.

$\textcircled{A} = \textcircled{A'}$: By assumption, $W^o = W^{\tilde{o}}$, thus we immediately have

$$\left(W^{\tilde{o}}F^{\tilde{o}}(s(z_u))\right)^T = \left(W^oF^{\tilde{o}}(s(z_u))\right)^T = W^o\left(F^{\tilde{o}}(s(z_u))\right)^{T_{23}}. \quad (72)$$

$\textcircled{B} = \textcircled{B'}$: Since $\alpha = \langle \mathbf{Q}, \mathbf{K}\rangle$, we compute $\mathbf{Q}$ and $\mathbf{K}$ for $\textcircled{B}$ and $\textcircled{B'}$ respectively. As for $\mathbf{Q}$, we have

$$\mathbf{Q}(s(f), z_u, s(z_v)) = \bigoplus_{o\in I(s(f))} W_Q^o(s(F))^o(z_u) = \bigoplus_{o\in I(s(f))} W_Q^o\left(F^{\tilde{o}}(s(z_u))\right)^T$$

$$= \bigoplus_{o\in I(s(f))} \left(W_Q^{\tilde{o}}F^{\tilde{o}}(s(z_u))\right)^T,$$

where the last equation holds because $W_Q^o = W_Q^{\tilde{o}}$ by assumption. In addition,

$$\mathbf{Q}(f, s(z_u), z_v) = \bigoplus_{o \in I(f)} W_Q^o F^o(s(z_u)).$$

As for $\mathbf{K}$, we have

$$\mathbf{K}(s(f), z_u, s(z_v)) = \bigoplus_{o \in I(s(f))} \sum_{i \in I(s(f))} \mathcal{W}_K^{o,i}(s(z_v) - z_u)\big(s(f)\big)^i(s(z_v))$$

$$= \bigoplus_{o \in I(s(f))} \Big( \sum_{i \in I(f)} \mathcal{W}_K^{\tilde{o},i}(z_v - s(z_u))f^i(z_v) \Big)^T,$$

where the last equation holds due to the constraint of radial function and Lemma. C.9, and we replace $i$ by $\tilde{i}$. In addition,

$$\mathbf{K}(f, s(z_u), z_v) = \bigoplus_{o \in I(f)} \sum_{i \in I(f)} \mathcal{W}_K^{o,i}(z_v - s(z_u))f^i(z_v).$$

We observe that $\mathbf{Q}(s(f), z_u, s(z_v))$ is just the transpose and re-ordering of each component of $\mathbf{Q}(f, s(z_u), z_v)$, and this is also true for $\mathbf{K}(s(f), z_u, s(z_v))$ and $\mathbf{K}(f, s(z_u), z_v)$, which suggests that their inner products are the same. Specifically, let $\mathbf{Q}^o = W_Q^o F^o(s(z_u))$ and $\mathbf{K}^o = \sum_{i \in I(f)} \mathcal{W}_K^{o,i}(z_v - s(z_u))f^i(z_v)$. We have

$$\alpha\big(s(f), z_u, s(z_v)\big) = \langle \bigoplus_{o \in I(s(f))} \big(\mathbf{Q}^{\tilde{o}}\big)^T, \bigoplus_{o \in I(s(f))} \big(\mathbf{K}^{\tilde{o}}\big)^T \rangle = \sum_{o \in I(s(f))} \langle \mathbf{Q}^{\tilde{o}}, \mathbf{K}^{\tilde{o}} \rangle = \sum_{o \in I(f)} \langle \mathbf{Q}^o, \mathbf{K}^o \rangle$$

and

$$\alpha\big(f, s(z_u), z_v\big) = \langle \bigoplus_{o \in I(f)} \mathbf{Q}^o, \bigoplus_{o \in I(f)} \mathbf{K}^o \rangle = \sum_{o \in I(f)} \langle \mathbf{Q}^o, \mathbf{K}^o \rangle,$$

which suggests that $\alpha(f, s(z_u), z_v) = \alpha(s(f), z_u, s(z_v))$.

$\textcircled{C} = \textcircled{C'}$: By assumption, the radial function satisfies $\varphi_{J_1,J_2}^{i,o}(\|z^1\|, \|z^2\|) = \varphi_{J_2,J_1}^{\tilde{i},\tilde{o}}(\|z^2\|, \|z^1\|)$ for all $i$, $o$, $J_1$, $J_2$, $z^1$ and $z^2$, thus $\mathcal{W}^{o,i}(z) = \big(\mathcal{W}^{\tilde{o},\tilde{i}}(s(z))\big)^{T_{12}T_{34}}$ according to Lemma. C.9. As a result,

$$\big(\mathcal{W}^{\tilde{o},i}(z_v - s(z_u))f^i(z_v)\big)^T = \big((\mathcal{W}^{o,\tilde{i}}(s(z_v) - z_u))^{T_{12}T_{34}} f^i(z_v)\big)^T = \mathcal{W}^{o,\tilde{i}}(s(z_v) - z_u)\big(f^i(z_v)\big)^T. \tag{73}$$

In summary, we have shown the $\mathbb{Z}/2\mathbb{Z}$-equivariance of a transformer layer (5), which is the first part of Prop. 5.2. We omit the proof of the second part of this proposition, *i.e.*, the equivariance of an Elu layer, as it is a simple extension of the proof of $\textcircled{A} = \textcircled{A'}$. $\qquad \square$

*Remark* C.10. The weight sharing technique in Prop. 5.2 reduces the number of learnable parameters of a $SE(3) \times SE(3)$-transformer by about half, thus it makes the model more efficient. In practice, the weight sharing technique can be achieved by sharing weights between different $o$, $i$ and $J$ directly, except for the symmetric case, *i.e.*, $o = \tilde{o}$, $i = \tilde{i}$ and $J_1 = J_2$, where the requirement of the radial function becomes

$$\varphi_{J_1,J_1}^{o,i}(\|z^1\|, \|z^2\|) = \varphi_{J_1,J_1}^{o,i}(\|z^2\|, \|z^1\|). \tag{74}$$

We represent the radial function in this case as $\varphi(x, y) = \frac{1}{2}(\phi(x, y) + \phi(y, x))$, where $\phi$ is a neural network, which guarantees the symmetry of $\varphi$.

### C.3.2 The complete-matching property

Consider the following complete matching problem.

**The complete matching problem** Given a pair of PCs $X$ and $Y$, where $Y$ is generated by a unknown rigidly transformation of $X$, *i.e.*, $Y = gX$ and $g \in SE(3)$ is unknown, how to infer $g$?

The complete matching problem is a prototype of the registration problem, *i.e.*, in the simplest case (no outlier, no noise, no partial visibility..), how to derive the relative transformation between two fully overlapped PCs? Despite its simplicity, this problem is non-trivial, because we do not know the correspondence between $X$ and $Y$, *i.e.*, Arun's method does not apply.

A surprising fact is that $g$ can be exactly recovered using a $SE(3)$-bi-equivariant and swap-equivariant assembly method. We state this fact formally as follows.

**Proposition C.11** (Complete-matching property)**.** *Let $\Phi$ be a swap-equivariant and $SE(3)$-bi-equivariant assembly method. Then*

$$\Phi(X, g_1 X)\Phi(X, X) = g_1, \tag{75}$$

*for arbitrary PC $X$ and $g_1 \in SE(3)$.*

*Proof.* First, since $\Phi$ is swap-equivariant, then $\Phi(X, X)\Phi(X, X) = I$. Then, since $\Phi$ is $SE(3)$-bi-equivariant, we have $\Phi(X, g_1 X)\Phi(X, X) = g_1\Phi(X, X)\Phi(X, X) = g_1$. $\square$

According to the main text, BITR is swap-equivariance (Prop. 5.2), thus, we can derive a concrete algorithm, called untrained BITR (U-BITR), for the complete matching problem:

$$\Phi_U(X, Y) = \Phi_B(X, Y)\Phi_B(X, X), \tag{76}$$

where $\Phi_B$ is BITR model. Prop. C.11 suggests that $\Phi_U$ solves the complete matching problem without training. We numerically evaluate this property in Appx. D.4, and leave the theoretical analysis of U-BITR to future research.

## C.4 The proof of Prop. 5.5

First, we have the following straightforward lemma for the kernel.

**Lemma C.12.** *If the radial function $\varphi : \mathbb{R} \times \mathbb{R} \to \mathbb{R}$ is a degree-$p$ function, then kernel $\mathcal{W}$ (5) satisfies*

$$\mathcal{W}(cz) = c^p \mathcal{W}(z), \quad \forall c \in \mathbb{R}_+. \tag{77}$$

Then we directly verify Prop. 5.5.

*The proof of Prop. 5.5.* Let $L$ be a transformer layer (5), $f$ be a degree-0 $\mathbb{R}_+$-equivariant input field, $\varphi_K$ be a degree-0 function. We seek to prove: 1) If $\varphi_V$ is a degree-0 function, then $c(L(f)) = L(c(f))$ and $L(f)$ is degree-0 $\mathbb{R}_+$-equivariant; 2) If $\varphi_V$ is a degree-1 function and $W = 0$, then $c(L(f)) = L(c(f))$ and $L(f)$ is degree-1 $\mathbb{R}_+$-equivariant.

To begin with, we expand $L(c(f))$ as

$$
\left(L\big(c(f)\big)\right)^{\boldsymbol{o}}(z_u) = W^{\boldsymbol{o}}(c(F))^{\boldsymbol{o}}(z_u) + \sum_{\substack{\boldsymbol{i} \\ z_v \in supp(c(f))}} \alpha(c(f), z_u, z_v)\mathcal{W}_V^{\boldsymbol{o,i}}(z_v - z_u)\big(c(f)\big)^{\boldsymbol{i}}(z_v)
$$

$$
= W^{\boldsymbol{o}}F^{\boldsymbol{o}}(c^{-1}z_u) + \sum_{\substack{\boldsymbol{i} \\ z_v \in supp(c(f))}} \alpha(c(f), z_u, z_v)\mathcal{W}_V^{\boldsymbol{o,i}}(z_v - z_u)f^{\boldsymbol{i}}(c^{-1}z_v)
$$

$$
= \underbrace{W^{\boldsymbol{o}}F^{\boldsymbol{o}}(c^{-1}z_u)}_{\textcircled{A}} + \sum_{\substack{\boldsymbol{i} \\ z_v \in supp(f)}} \underbrace{\alpha(c(f), z_u, c(z_v))}_{\textcircled{B}}\underbrace{\mathcal{W}_V^{\boldsymbol{o,i}}(c(z_v) - z_u)f^{\boldsymbol{i}}(z_v)}_{\textcircled{C}},
$$

and expand $c(L(f))$ as

$$
\left(c\big(L(f)\big)\right)^{\boldsymbol{o}}(z_u) = c^p \left(\big(L(f)\big)\right)^{\boldsymbol{o}}(c^{-1}z_u)
$$

$$
= c^p \left(\underbrace{W^{\boldsymbol{o}}F^{\boldsymbol{o}}(c^{-1}z_u)}_{\textcircled{A'}} + \sum_{\substack{\boldsymbol{i} \\ z_v \in supp(f)}} \underbrace{\alpha(f, c^{-1}z_u, z_v)}_{\textcircled{B'}}\underbrace{\mathcal{W}_V^{\boldsymbol{o,i}}(z_v - c^{-1}z_u)f^{\boldsymbol{i}}(z_v)}_{\textcircled{C'}}\right).
$$

We first point out that $\widehat{B} = \widehat{B'}$, because

$$\mathbf{K}(c(f), z_u, c(z_v)) = \bigoplus_o \sum_i \mathcal{W}_K^{o,i}(c(z_v) - z_u)(c(f))^i(z_v) = \bigoplus_o \sum_i \mathcal{W}_K^{o,i}(z_v - c^{-1}(z_u))f^i(c^{-1}z_v)$$
$$= \mathbf{K}(f, c^{-1}z_u, z_v),$$

where the second equation holds because $\varphi_K$ is a degree-0 function by assumption, and Lemma. C.12 claims that the corresponding kernel $\mathcal{W}_K$ is scale-invariant. In addition,

$$\mathbf{Q}(c(f), z_u, c(z_v)) = \bigoplus_o W_Q^o(c(F))^o(z_u) = \bigoplus_o W_Q^o F^o(c^{-1}z_u) = \mathbf{Q}(f, c^{-1}z_u, z_v).$$

In other words, both $\mathbf{Q}$ and $\mathbf{K}$ are scale-invariant. Thus, we conclude that the attention is also scale-invariant:

$$\alpha(c(f), z_u, c(z_v)) = \langle \mathbf{Q}(c(f), z_u, c(z_v)), \mathbf{K}(c(f), z_u, c(z_v)) \rangle$$
$$= \langle \mathbf{K}(f, c^{-1}z_u, z_v), \mathbf{Q}(f, c^{-1}z_u, z_v) \rangle$$
$$= \alpha(f, c^{-1}z_u, z_v).$$

Now we discuss these two situations:

1) If $\varphi_V$ is a degree-1 function and $W = 0$, then

$$c\mathcal{W}_V^{o,i}(z_v - c^{-1}z_u) = \mathcal{W}_V^{o,i}(c(z_v) - z_u)$$

according to Lemma. C.12, and $\widehat{A} = \widehat{A'} = 0$. In other words, $L(c(f))(z) = c^1 L(f)(c^{-1}z)$, where the RHS is the action of $c$ on the degree-1 $\mathbb{R}_+$-equivariant field, thus we have $L(c(f)) = c(L(f))$, and $L(f)$ is degree-1 $\mathbb{R}_+$-equivariant.

2) If $\varphi_V$ is a degree-0 function, then

$$\mathcal{W}_V^{o,i}(z_v - c^{-1}z_u) = \mathcal{W}_V^{o,i}(c(z_v) - z_u)$$

according to Lemma. C.12. Thus, $L(c(f))(z) = L(f)(c^{-1}z)$, where the RHS is the action of $c$ on the degree-0 $\mathbb{R}_+$-equivariant field, thus we have $L(c(f)) = c(L(f))$, and $L(f)$ is degree-0 $\mathbb{R}_+$-equivariant.

In summary, we have shown that the equivariance of the transformer layer. We omit the proof of the equivariance of the Elu layer as it can be proved similarly. $\qquad \square$

We can now construct a special structure of BITR for producing degree-1 $\mathbb{R}_+$-equivariant output:

$$deg\text{-}0 \;\rightarrow\; deg\text{-}0 \;\rightarrow\; \cdots \rightarrow\; deg\text{-}0 \;\rightarrow\; deg\text{-}1 \tag{78}$$

In other words, all tensor fields are degree-0 $\mathbb{R}_+$-equivariant, except for the final output which is degree-1 $\mathbb{R}_+$-equivariant. More specifically, we consider two types of layers, *i.e.*, *deg*-0 $\rightarrow$ *deg*-0 and *deg*-0 $\rightarrow$ *deg*-1.

Prop. 5.5 suggests that a transformer layer with degree-$p$ radial function can generate a degree-$p$ $\mathbb{R}_+$-equivariant field when the input field is degree-0 $\mathbb{R}_+$-equivariant (Elu layers do not influence the $\mathbb{R}_+$-equivariance). Therefore, we only need to consider degree-0 and degree-1 radial functions for our purpose. In practice, we represent degree-1 functions using neural networks consisting of linear and relu [1] layers, and we represent degree-0 functions using linear, relu and layer normalization [3] layers. Specifically, We represent degree-0 functions as neural network $\phi_0$:

$$\phi_0 : \quad Linear \rightarrow Relu \rightarrow LayerNorm \rightarrow Linear \rightarrow Relu \rightarrow LayerNorm, \tag{79}$$

and represent degree 1-functions neural network $\phi_1$:

$$\phi_1 : \quad Linear \rightarrow Relu \rightarrow Linear \rightarrow Relu. \tag{80}$$

*Remark* C.13. There are two major difficulties in developing a general scale-equivariance theory for BITR. First, the attention term is not scale-invariant due to the existence of the soft-max operation, *i.e.*, *SoftMax*$(A) \neq$ *SoftMax*$(cA)$ for vector $A$ and $c \in \mathbb{R}_+$. That is the reason why we only accept degree-0 $\mathbb{R}_+$-equivariant input in our current theory. Second, when $p \notin \{0, 1\}$, we are not aware of any general way to represent degree-$p$ functions using neural networks.

# D More experiment results

## D.1 More training details

We run all experiments using a Nvidia T4 GPU card with 16G memory. The batch size is set to the largest possible value that can be fitted into the GPU memory. We set $bs = 16$ for the airplane dataset, and $bs = 4$ for the wine bottle dataset. We train BITR until the validation loss does not decrease. For the airplane dataset, we train BITR 10000 epochs, and the training time is about 8 days when $s = 0.7$. For the wine bottle dataset, we train BITR 1000 epochs, and the training time is about 12 hours. The FLOPS is $14.5G$ in a forward pass (including the computation of harmonic functions), and the model contains $0.17M$ parameters.

For Sec. 6.3.1 and 6.3.2, we use the normal vector computed by Open3D [46] as the input feature of BITR. The airplane dataset used in Sec. 6.3.1 contains 715 random training samples and 103 random test samples. The wine bottle dataset used in Sec. 6.4 contains 331 training and 41 test samples. We adopt the few-shot learning setting in the manipulation tasks in Sec. 6.6: we use 30 training and 5 test samples for mug-hanging; we use 40 training samples and 10 test samples for bowl-placing.

## D.2 More results of Sec. 6.2

We quantitatively verify the equivariance of BITR according to Def. 3.1. Specifically, we compute

$$\Delta_{bi} = \|\Phi_B(g_1 X, g_2 Y) - g_2 \Phi_B(X, Y) g_1^{-1}\|_F, \tag{81}$$

$$\Delta_{swap} = \|\Phi_B(Y, X) - (\Phi_B(X, Y))^{-1}\|_F, \tag{82}$$

$$\Delta_{scale} = \|r_B(cX, cY) - r_B(X, Y)\|_F + \|t_B(cX, cY) - ct_B(X, Y)\|_2, \tag{83}$$

to verify the $SE(3)$-bi-equivariance, swap-equivariance and scale-equivariance of BITR, where $\Phi_B$ represent the BITR model, $(r_B(\cdot), t_B(\cdot)) = \Phi_B(\cdot)$ are the output of BITR, and all $g = (r, t) \in SE(3)$ are written as

$$g = \begin{bmatrix} r & t \\ 0 & 1 \end{bmatrix} \in \mathbb{R}^{4,4}. \tag{84}$$

Note that if BITR is perfectly equivariant, these three errors should always be 0.

The quantitative results of the experiment are summarized in Tab. 3, where we can see that all errors are below the numerical precision of float numbers, *i.e.*, less than $1e^{-5}$. The results suggest that BITR is indeed $SE(3)$-bi-equivariant, swap-equivariant and scale-equivariant.

Table 3: Verification of the equivariance of BITR.

| $\Delta_{bi}$ | $\Delta_{swap}$ | $\Delta_{scale}$ |
|---|---|---|
| $5e^{-6}$ | $2e^{-7}$ | $5e^{-7}$ |

## D.3 Ablation study

To show the practical effectiveness of our theory on scale and swap equivariances, we consider an ablation study. We use the same data as in Sec. 6.2, and remove the weight sharing technique in Sec. 5.1 to break swap-equivariance, and force $\varphi_V$ in all layers to be a degree-1 function to break the scale-equivariance.

We evaluate the trained model on 100 test samples, and report the mean and standard deviation of $\Delta r$ in Tab. 4, where we can see that removing an equivariance of the model leads to the failure in the corresponding test case, which is consistent with our theory.

Table 4: Ablation study of scale and swap equivariances. We report mean and std of $\Delta r$

| | original | rigidly perturbed | swapped | scaled |
|---|---|---|---|---|
| BITR (full model) | (17.1, 6.0) | (17.1, 6.0) | (17.1, 6.0) | (17.1, 6.0) |
| BITR (w/o swap) | (15.3, 7.0) | (15.3, 7.0) | (70.3, 20.0) | (15.3, 7.0) |
| BITR (w/o scale) | (15.0, 8.0) | (15.0, 8.0) | (15.0, 8.0) | (125.4, 15.0) |

### D.4 Evaluation of the complete-matching property

This experiment numerically evaluates the robustness of the complete-matching property (Prop. C.11) against resampling, noise and partial visibility. We first sample $X$ and $Y$ of size $1024$ from the bunny shape, and a random $g \in SE(3)$, then we use a random initialized U-BITR to match $X$ to $gY$. We consider different settings: 1) $X$ and $Y$ are exactly the same; 2) $X$ and $Y$ are different random samples; 3) Gaussian noise of std $0.01$ is added to $X$ and $Y$; 4) Ratio $s$ of $X$ and $Y$ is kept by cropping using a random plane.

We repeat the experiment 3 times, and report the results of U-BITR in Tab. 5. We observe that the transformation is perfectly recovered when $X = Y$, which is consistent with the complete-matching property. Meanwhile, cropping the PCs leads to large decrease of the accuracy, while noise and resampling have less effect. This is consistent with our expectation because cropping the PCs has larger effect on the shape of PCs. We provide qualitative results in Fig. 7.

Table 5: Results of complete matching using U-BITR.

|  | $\Delta r$ | $\Delta t$ |
|---|---|---|
| $X = Y$ | 0.0 (0.0) | 0.0 (0.0) |
| Resampled | 20.7 (16.4) | 0.18 (0.16) |
| Noisy | 5.0 (2.8) | 0.02 (0.008) |
| Cropping $s = 0.9$ | 99.1 (40.49) | 0.58 (0.4) |

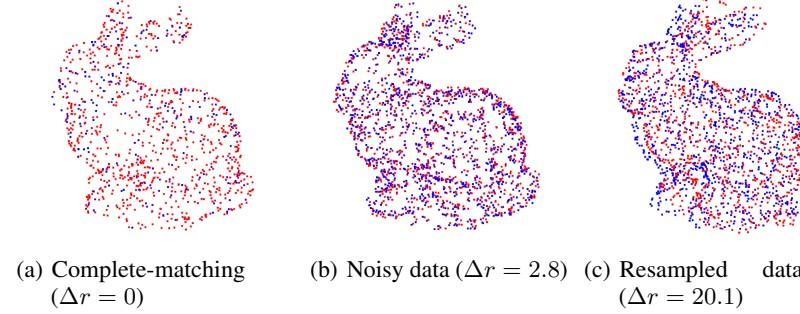

(a) Complete-matching
($\Delta r = 0$)

(b) Noisy data ($\Delta r = 2.8$)

(c) Resampled data
($\Delta r = 20.1$)

Figure 7: Qualitative results of U-BITR on complete matching. The blue and red PCs are the reference and the transformed source PCs respectively. Note that U-BITR is only random initialized (not trained).

### D.5 More results of Sec. 6.3.1

We report the training process of BITR in Fig. 8, where we can see that the loss value, $\Delta r$ and $\Delta t$ gradually decrease during training as expected.

Some qualitative results of BITR are presented in Fig. 9. We represent the input PCs using light colors, and represent the 32 learned key points using dark colors and large points. As we explained in Sec. 4.3, the key points are in the convex hull of the input PCs, and they are NOT a subset of the input PC. In addition, as can be seen, the key points of the inputs do not overlap.

### D.6 More results of Sec. 6.3.2

We generate the raw training samples by sampling motorbike and car shapes from the training set of ShapeNet. Then we centralize them, and move the car by $[0, 0, 1]$. Note the shapes in ShapeNet are already pre-aligned. The test samples are generated from the test set of ShapeNet in the same way.

We repeat the test process 3 times, and report the results in Fig. 10. We observe that BITR achieves lower rotation error than LEV, and their translation errors are comparable. Meanwhile, NSM fails in this experiment. Note that we do not report the results of registration methods, because their loss functions are undefined due to the lack of correspondence.

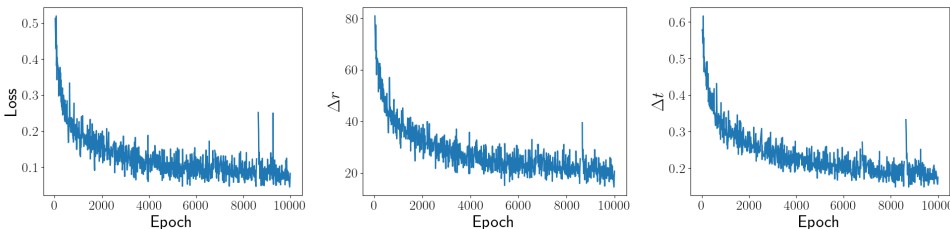

Figure 8: The training process of BITR on the airplane dataset with $s = 0.4$. All metrics are measured on the validation set.

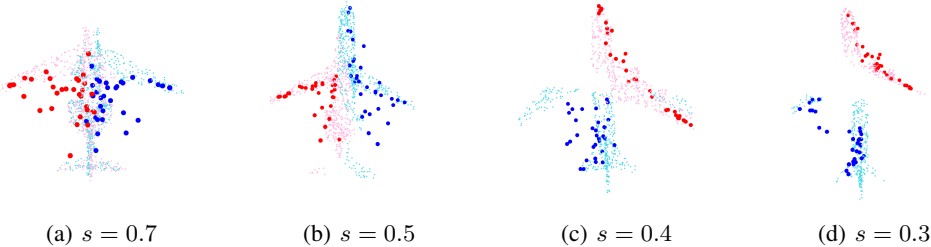

(a) $s = 0.7$      (b) $s = 0.5$      (c) $s = 0.4$      (d) $s = 0.3$

Figure 9: The PC registration results of BITR on the airplane dataset. The input PCs are represented using light colors, and the learned key points are represented using dark and large points.

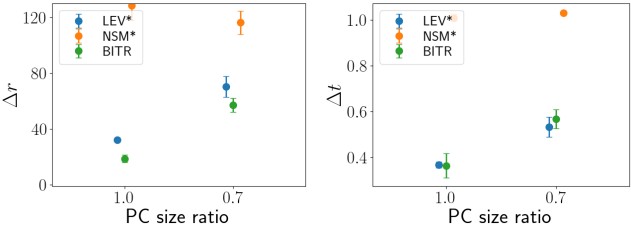

Figure 10: Assembly results of car and motorbike. $*$ denotes methods which require the true canonical poses of the input PCs.

## D.7   More results of Sec. 6.4

For BITR, we first obtain raw PCs by applying grid sampling with grid size $0.005$ to the shape, and then randomly sample $5\%$ points from the raw PCs as the training and test samples. The sizes of the resulting PCs are around $1000$, which is close to the data used in the baseline methods. The data is pre-processed following [38] for the baseline methods.

The random sampling process in our method causes the randomness of test error. We quantify the randomness by evaluating on the test set 3 times, and report the mean and std of the errors in Tab. 1.

Fig. 11 presents $5$ examples of reassembling wine bottle fragments, where we observe that the proposed BITR can reassemble most of the shapes correctly, while the baseline methods generally have difficulty predicting the correct rotations.

## D.8   More results of Sec. 6.5

We preprocess the 7Scenes dataset by applying grid sampling with grid size $0.1$. The training and test sets contain $278$ and $59$ samples.

We consider the $5$ outdoor scenes in the ASL dataset: mountain, winter, summer, wood autumn, wood summer. We preprocess all data by applying grid sampling with grid size $0.9$. We arbitrarily rotate and translate all data, and train BITR to align the $i$-th frame to the $(i + 2)$-th frame. We use the first

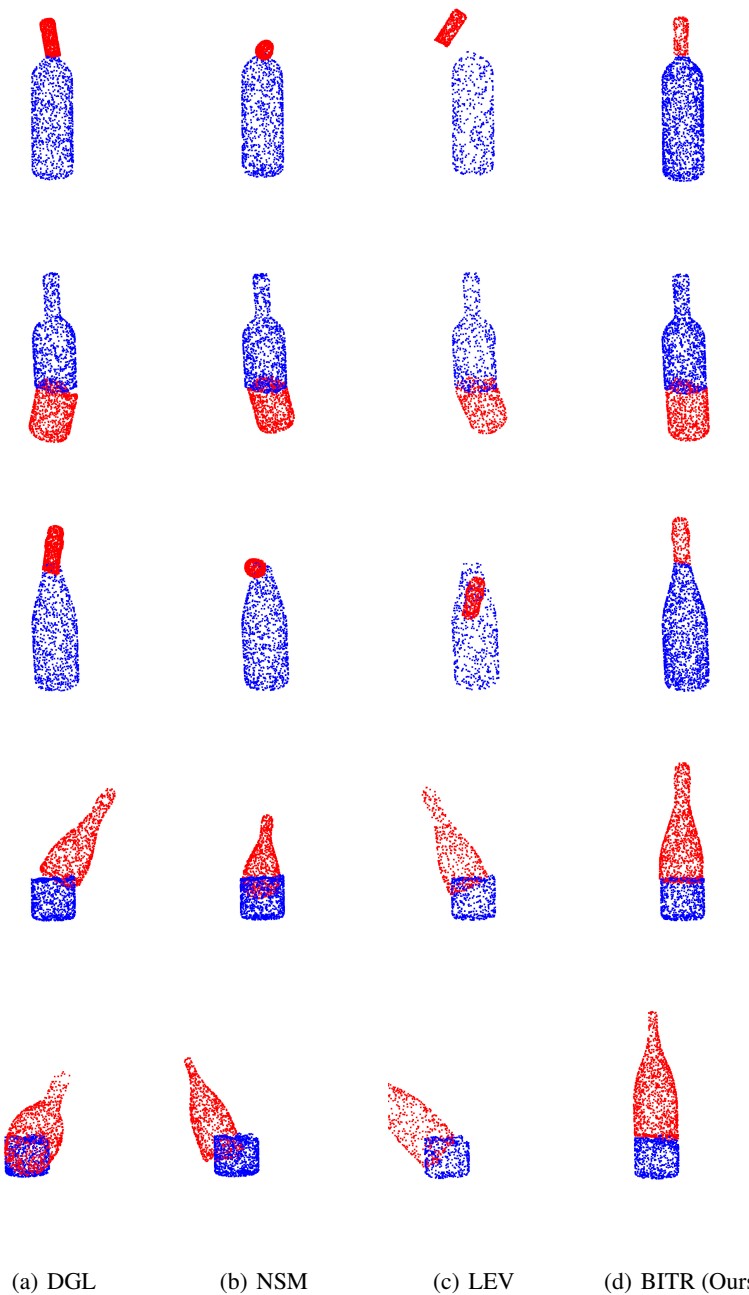

(a) DGL      (b) NSM      (c) LEV      (d) BITR (Ours)

Figure 11: Results of reassembling wine bottle fragments. We compare the proposed BITR with DGL [44], NSM [7] and LEV [38]. Zoom in to see the details.

20 frames as the training set, and the other frames as the test set. This leads to a training set of size 105, and a test set of size 48

We report the results in Tab. 6. Our observation is consistent of that in the 7Scenes dataset: the result BITR is close to the optimum, thus a refinement can lead to improved results. Note that BITR+ICP outperforms all baselines in this task. In addition, GEO causes the out-of-memory error on our $16G$ GPU. An assembly result (wood summer) is presented in Fig. 12.

Table 6: Results on the outdoor scenes of ASL. We report mean and std of $\Delta r$ and $\Delta t$.

|  | $\Delta r$ | $\Delta t$ |
|---|---|---|
| ICP | 73.2 (8.0) | 9.4 (1.0) |
| OMN | 110.0 (5.0) | 2.0 (1.0) |
| GEO | − | − |
| ROI | 16.7 (0.0) | 1.5 (0.0) |
| BITR (Ours) | 10.6 (0.0) | 1.4 (0.0) |
| BITR+ICP (Ours) | 0.7 (0.0) | 0.8 (0.0) |

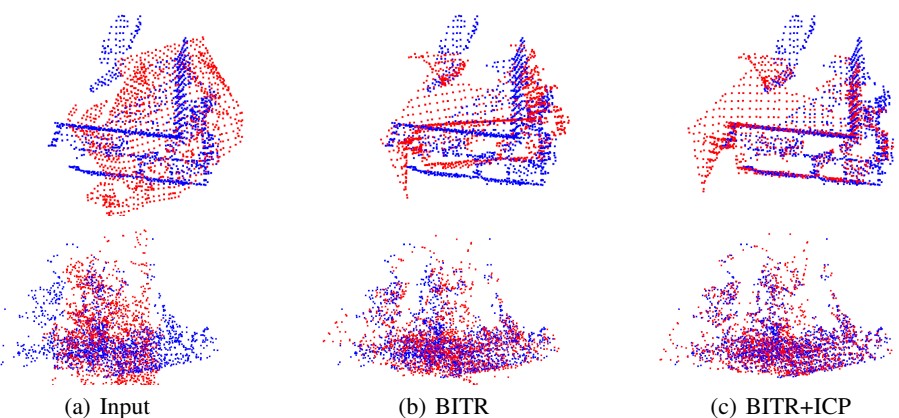

(a) Input        (b) BITR        (c) BITR+ICP

Figure 12: An assembly result of BITR on 7Scenes (1-st row) and ASL (2-nd row). BITR can produce results that are close to the optimum, and a ICP refinement leads to the improved results.

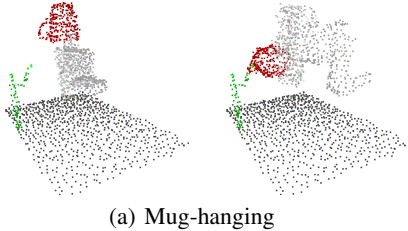

(a) Mug-hanging

Figure 13: The result of BITR on mug-hanging.

### D.9 More results of Sec. 6.6

All data in this experiment is generated using PyBullet [8], where the objects in training and test sets are different in shape and position.

Note that we do not report the quantitative results for this experiment because the metric $(\Delta r, \Delta t)$ is ambiguous due to the non-uniqueness of the correct solution. For example, for a correct bowl-placing result, the bowl is allowed to rotate horizontally in the plate, so the metric $\Delta r$ can be very large even for this correct assembly. A suitable metric is the success rate of the manipulation in physical hardware experiments as in [27], but measuring this metric is beyond the scope of this work. In addition, the assembly methods such as NSM [7] and LEV [38] are not applicable because the canonical pose is not known, and we do not report the result of any registration method because the correspondence does not exist. We present a result of BITR on mug-hanging in Fig. 13.

## E    Limitations and future research directions

In our current implement, we have accelerated most of the layers of BITR using the "scatter" function [24]. However, BITR is still relatively slow due to the independent computation of convolutional

kernels, *i.e.*, the harmonic function and the independent multiplication of radial and angular component in each degree. This is reflected by a low GPU utility ratio (about $20\%$). We expect to see a large speed gain (about $\times 5$) if the above-mentioned computation is implemented using CUDA kernel, *i.e.*, GPU utility ratio can be close to $100\%$. On the other hand, a rotation-based technique was recently introduced to reduce the computation cost of SE(3)-equivariant networks [21]. A promising future direction is to extend this technique to BITR.

To explain the limitation of BITR in handling symmetric PCs, we consider the following example. For a pair of PCs $X$ and $Y$, if there exists a non-identity rigid transformation $g_2$ such that $g_2 Y = Y$, and $g \in SE(3)$ is a proper transformation that assembles $gX$ to $Y$, then $g_2 g$ is an equally good transformation that gives the same assembly result. In other words, the optimal transformation $g$ is non-unique, which cannot be modelled by BITR. Actually, we notice that the result of BITR is undefined in this case. Specifically, let $\tilde{g} = \Phi_B(X, Y)$, we have

$$\tilde{g} = \Phi_B(X, Y) = \Phi_B(X, g_2 Y) = g_2 \Phi_B(X, Y) = g_2 \tilde{g} \tag{85}$$

due to $SE(3)$-bi-equivariance. Then we have $g_2 = 1$, which contradicts the assumption.

To address this issue, we plan to extend BITR to a generative model in future research, *i.e.*, it should assign a likelihood value to each prediction. For the example considered above, it should assign equal probability to $g$ and $g_2 g$.

Apart from the above-mentioned two limitations, there are several directions for future investigation. First, it is important to generalize BITR to multi-PC assembly tasks where more than 2 PCs are considered [14, 38]. Second, we expect the U-BITR model to be useful in self-supervised 3D shape retrieval or detection models, such as those in image object detection/retrieval [19].

