# OpenReview forum: "SE(3)-bi-equivariant Transformers for Point Cloud Assembly"
_NeurIPS.cc/2024/Conference — NeurIPS 2024 poster_

### Official Review · Reviewer_1793 · 2024-06-23

**Soundness:** 3
**Presentation:** 3
**Contribution:** 2
**Rating:** 5
**Confidence:** 4

**Summary:**

This paper presents an end-to-end framework for pointcloud correspondence and relative pose estimation with bi-equivariance to per-part poses. The framework is also equivariant to scaling and swapping part orders. It also designes a transformer architecture with $\mathrm{SE}(3)\times\mathrm{SE}(3)$-equivariance. The effectiveness of the framework is shown with experiments on the standard ShapeNet benchmark as well as some other applications.

**Strengths:**

- The problem studied in this paper is well-suited for equivariance, and the paper provides a decent problem formulation with clear explanations.
- The network architecture in the paper is well-designed, with the bi-equivariant transformer, and the module with the classic pose matching algorithm.
- The experiments are extensive in the sense that they not only show standard comparisons on ShapeNet, but also have more studies with different setups and other applications.

**Weaknesses:**

- To my understanding, "shape assembly" usually refers to assembling parts of a shape that exactly matches their boundaries (like jigsaw), and usually with more than just two parts. I feel "shape registration" is a more proper description of the task in the paper.
- Following my previous comment, I think the method should also be compared to prior works on shape registration, including classic optimization-based methods like ICP, and more recent learning-based methods such as [1, 2] below.
	- A minor point: to compare to these works and show the advantages brought by equivariance, it may be a good setting to compare their data efficiency. I believe that with less training data or less pose augmentation, the proposed framework can have less performance drop than these non-equivariant prior works.
- [3] is also a related work, especially to the robot manipulation task in Sec. 6.6 (and Suppl. Sec. D.9).

[1] Wang, Y., & Solomon, J. M. (2019). Deep closest point: Learning representations for point cloud registration. In Proceedings of the IEEE/CVF international conference on computer vision (pp. 3523-3532).

[2] Wang, Y., & Solomon, J. M. (2019). Prnet: Self-supervised learning for partial-to-partial registration. Advances in neural information processing systems, 32.

[3] Ryu, H., Kim, J., An, H., Chang, J., Seo, J., Kim, T., ... & Horowitz, R. (2024). Diffusion-edfs: Bi-equivariant denoising generative modeling on se (3) for visual robotic manipulation. In Proceedings of the IEEE/CVF Conference on Computer Vision and Pattern Recognition (pp. 18007-18018).

**Questions:**

I don't have questions about the method. But I would really like to see experimental comparisons with the pointcloud registration works.

**Limitations:**

Limitations are well-discussed in the paper.

---

> ### Author Rebuttal · Authors · 2024-08-05
>
> We thank the reviewer for the time and effort. We address the concerns below.
>
>
> 1.  I feel "shape registration" is a more proper description of the task in the paper.
>
> We use the word "assembly" following [7], where two pieces of point cloud are matched together. We avoid using the word "registration" because we are afraid this word will cause some confusion: registration is a classic task which seeks to align overlapped point clouds using correspondence, so aligning non-overlapped point clouds seems not well-defined for this task.
>
>
> 2.  Should be compared to prior works on shape registration, including classic optimization-based methods like ICP, and more recent learning-based methods such as [1, 2] below.
>
> We have already included two state-of-the-art registration methods [21] and [39] as baselines (they are more recent than the papers mentioned by the reviewer). These two methods are learning based and have an optimization-type finetuning process. ICP is not compared in the main text because it generally fails when the pose is randomly initialized (e.g. when the initial rotation error is larger than 45 degrees). We have included ICP in table 5 and table 6 in the appendix for clarity.
>
> 3.  with less training data or less pose augmentation, the proposed framework can have less performance drop than these non-equivariant prior works...
>
> We showed that one important advantage of our method brought by equivariance is that it can be used when correspondence does not exist (Fig.4 when $s < 0.5$). Of course, you are correct about the data efficiency, because we do not use pose augmentations as noted in line 296 (they have no influence on our method), but they are necessary for learning-based methods like [21].
>
> 4. [3] is also a related work, especially to the robot manipulation task in Sec. 6.6 (and Suppl. Sec. D.9).
>
> Yes, this is a related paper. We have now cite this paper, but we are not able to conduct quantitative comparisons, because the correct solution is not unique in this task (as noted in line 889). For example, for a correct assembly, the bowl is allowed to rotate horizontally in the plate, so $\Delta r$ can be very large even for the correct assembly. A proper metric is the success rate of the manipulation in physical hardware experiments as in [3], but measuring this metric is beyond the scope of this work.

---

### Official Review · Reviewer_1H6H · 2024-07-11

**Soundness:** 3
**Presentation:** 3
**Contribution:** 3
**Rating:** 5
**Confidence:** 3

**Summary:**

This paper introduces a new network to solve the point cloud assembly task, where a 3D transformation is predicted to “align” two point clouds. The proposed network, BTIR, is designed to enforce the symmetries of the point cloud assembly task into the network layers. Specifically, it enforces SE(3) bi equivariance, scale equivariance and swap equivariance through weight constraints. The network is evaluated on several point cloud assembly tasks. It significantly outperforms existing equivariant methods in some settings.

**Strengths:**

- The paper is well-written. Motivating the idea as a learnable extension of Arun’s method was intuitive and easy to follow.
- The paper introduces a SE(3) biequivariant network, BTIR, to solve point cloud assembly tasks. The network is composed of novel SE(3) biequivariant group convolution layers.
- The paper provides weight constraints that enforce scale and swap equivariance. An ablation study is performed that demonstrates the added generalization provided by these constraints.
- The paper includes several PC alignment and assembly experiments where the proposed BTIR network is shown to perform well.

**Weaknesses:**

- There are several works from the robotic manipulation community that are related to BTIR. These should be discusses in RW and perhaps included as baselines, especially the bowl placing and mug hanging tasks. [1,2]
- The definition of point cloud assembly in the introduction is a bit imprecise. What does it mean for two non-overlapped point clouds to be “aligned”, when the points themselves will not overlap? Presumably what aligned means depends on who generates the dataset.
- Some of the writing is inaccurate or unclear. On page 2, “[correspondence based methods] are often sensitive to initial positions of PCs”; most correspondence based approaches use Aruns method which is not sensitive to initial poses. On page 2, “SE(3)-bi-equivariance does not rely on correspondence”; do you mean your BTIR method does not?
- The results on real data and visual manipulation are of interest and should be included in more detail in the main paper. They are more challenging and would better demonstrate that the proposed method is generally effective. Also, more discussion should be included about these results. For instance, why is it that BTIR without ICP is relatively bad on 7Scenes but better on outdoor scenes of ASL?
- The experimental results are somewhat limited. More baselines should be included (point cloud alignment is a very well studied problem) and more challenging or varied datasets used. For instance, compare performance across multiple shapenet classes rather than just the wine bottle. Very little space in the paper is currently dedicated to the experiments and discussion. Given BTIR is more computationally heavy than other methods, it should be clear from the experiments that it is robust and effective enough to motivate its use.

[1] Pan, Chuer, et al. "Tax-pose: Task-specific cross-pose estimation for robot manipulation." Conference on Robot Learning. PMLR, 2023.
[2] Ryu, Hyunwoo, et al. "Diffusion-edfs: Bi-equivariant denoising generative modeling on se (3) for visual robotic manipulation." Proceedings of the IEEE/CVF Conference on Computer Vision and Pattern Recognition. 2024.

**Questions:**

- In the limitations, it is mentioned that the method is deterministic. How does it solve the wine bottle task in Table 1? Is the loss computed up to the to next symmetry or is the symmetry broken in the loss?
- It is not clear whether the method is limited to type 0 and 1 features or if they were only mentioned for simplicity. Do the layers introduced in the paper work for higher order features ? Is there a reason other than computational effort that higher order features are not used?
- It should be mentioned that the enforcing scale equivariance on type 1 features was explored in another paper [1].
- Is this the first instance of biequivariant filters being used to process two inputs simultaneously? If so this should be stated more clearly as it has many interesting possibly applications even if it is more computationally costly.

[1] Yang, Jingyun, et al. "Equivact: Sim (3)-equivariant visuomotor policies beyond rigid object manipulation." arXiv preprint arXiv:2310.16050 (2023).

**Limitations:**

Limitations are discussed in Appendix E. If possible these limitations should be included in the main paper if only briefly.

---

> ### Author Rebuttal · Authors · 2024-08-05
>
> We thank the reviewer for the time and effort. We address the concerns below.
>
> 1. Robotic manipulation papers including Diffusion-edfs, Tax-pose and Sim3
>
> Yes. Those are related papers. We have now cited them, but as noted in line 889, we are not able to conduct quantitative comparisons, because the correct solution is not unique in this task. For example, for a correct assembly, the bowl is allowed to rotate horizontally in the plate, so $\Delta r$ can be very large even for the correct assembly. A proper metric is the success rate of the manipulation in physical hardware experiments as in [1,2], but measuring this metric is beyond the scope of this work.
>
> We have now added the following sentence (bold) in line 88: "... modelling 3D
> data, **and recently they have been used for robotic manipulation task [26 , 27, 38]** ". Since the Tax-pose paper does not use equivariant network, we feel that it is more suitable to mention it in line 16 (robotics [27, 20]) in our introduction.  (27 is the Diffusion-edfs paper and 20 is the Tax-pose paper, and 38 is the Sim3 paper)
>
> 2. The definition of point cloud assembly... is a bit imprecise. .. Presumably what aligned means depends on who generates the dataset.
>
> Yes. Depending on the specific dataset, it can mean reconstructing the shape, robotic manipulation, or even protein binding. We feel the word "align" is general enough to cover these meanings, and we provided some examples and citations in Line 16 to make the meanings concrete.
>
> 3. ... most correspondence-based approaches use Aruns method which is not sensitive to initial poses.
>
> Although Arun's method is robust to initial poses, most of the correspondence-based methods are sensitive because the features they use are not invariant (or equivariant). We mentioned this in line 69 in Sec.2.
>
> 4. ...“SE(3)-bi-equivariance does not rely on correspondence”; do you mean your BTIR method does not?
>
> Sorry for the confusion. We mean the SE(3)-bi-equivariance prior (the equation in definition 3.1) does not rely on correspondence. We have now made this sentence clearer (the added content is marked bold): "$SE(3)$-bi-equivariance **prior** does not rely on correspondence, i.e., it can **be used to** handle PCs with no correspondence."
>
> 5. The results on real data and visual manipulation...should be included in more detail in the main paper... Also, more discussion should be included.. why is it that BTIR without ICP is relatively bad on 7Scenes but better on outdoor scenes of ASL?
>
> We agree that the experiments are important, but due to the 9-page limit of the main text, we have to place them in the appendix and refer to them in the main text to keep the overall structure complete.
>
> On the other hand, we have now made it clearer that BITR+OT should be compared with GEO and ROI, because they also include an OT-type refinement process. The result of BITR is used to show that the model can generate results that are close to the optimum (the errors can vary in different datasets).
>
> Specifically, we re-organize the paragraph starting at line 848 as follows:
> "We report the results in Tab. 5. We observe that BITR can produce results that are close to the optimum ($\Delta r \approx 25$) from a random initialization ($\Delta r \in U [0, 180]$), and extra refinements like ICP and OT can further improve the results ($\Delta r \approx 10$). This observation is consistent with that in Sec. 6.3.1. In particular, BITR with the OT refinement is comparable with GEO and ROI, which use highly complicated features specifically designed for registration tasks and an OT-like refinement process. On the other hand, ICP and OMN fail in this task due to their sensitivity to initial positions. An example of an assembly result of BITR is presented in Fig. 11."
>
> 6. ...More baselines should be included ... and more challenging or varied datasets used. For instance, compare performance across multiple shapenet classes...
>
> - As for the registration baselines, we have included the state-of-the-art methods GEO [21] and ROI [39] as two strong baselines. We also considered the classic method ICP [41], and a recent correspondence-free method OMN [36] in Sec 6.5 for completeness. We think the comparisons with these methods are sufficient to support the argument that our method is at least comparable with the existing registration methods.
>
> - On the other hand, as for the dataset, we applied our method to different types of datasets, including indoor (7Scene), outdoor (ASL), objects (ShapeNet), and manipulation. We think these experiments can show the versatility of our method in practice. However, we are not able to train on larger datasets due to high computational cost as discussed in Appx.E. For example, training on one class of ShapeNet takes several days as noted in Appx.D.1, so we do not have enough time to train on multiple classes at this moment. We leave the task of scaling BITR to future research. Probably this can be done with the aid of the techniques mentioned by reviewer fq5D.
>
> 7. ...the method is deterministic. How does it solve the wine bottle task in Table 1?
>
> The input pieces of a bottle are not symmetric (there is no restriction on the complete bottle shape), so our method can be used without difficulty. Actually, as noted in line 910, we do not need to worry about the symmetry problems for the Lidar type data because strict symmetry never exists due to noise. (Symmetry is a problem when modelling molecules, such as a benzene ring, which is strictly symmetric.)
>
>
>
> 8. About the degree of features.
>
> Our layer works for higher order features. We did not use higher order features due to efficiency consideration. We will add a sentence below Eqn 12 to make this clearer: "here we only include degree-0 and degree-1 features, and higher degree features can be used similarily"
>
> 9. Is this the first instance of bi-equivariant filters being used to process two inputs simultaneously?
>
> Yes. We have stated it in line 92.

---

> > ### Comment · Reviewer_1H6H · 2024-08-08
> >
> > Thank you for the response.
> >
> > Re: 7, I am still confused.  In Figure 10, it appears that most of the fragments are symmetric about the up-axis, so doesn't that mean that many possible transformations produce good alignment?
> >
> > Re: 9, yes I see it now.  This is an important contribution to the field, and I think it should be restated clearly in the contribution bullets as well.

---

> > > ### Author Response · Authors · 2024-08-09
> > >
> > > Thanks for your reply.
> > >
> > > 1. The fragments are not symmetric (If they were, there would be many possible solutions). These fragments are from the BB dataset, which are generated by simulating the physical process of breaking the objects. So they are not likely to be symmetric (the broken surfaces can be non-regular). This can be seen by zooming in Fig.10.
> > >
> > > 2. Agree. We have now added a sentence in the first bullet point in the introduction section. "...In addition,
> > >     the $SE(3) \times SE(3)$-transformer used in BITR is the first $SE(3) \times SE(3)$-equivariant steerable network to the best of our knowledge."

---

### Official Review · Reviewer_fq5D · 2024-07-12

**Soundness:** 4
**Presentation:** 2
**Contribution:** 4
**Rating:** 9
**Confidence:** 4

**Summary:**

This paper proposes an SE(3)-bi-equivariant approach for point cloud assembly, addressing the difficulty of assembling non-overlapping point clouds where traditional correspondence matching methods struggle. The proposed BITR (BI-equivariant TRansformer) solves this problem by exploiting the symmetry of the point cloud assembly task as a powerful inductive bias, instead of relying on point correspondence.

Specifically, BITR first merges two distinct 3D point clouds into a 6D point cloud with features that are the tensor product (also known as the Kronecker product or outer product) representation of two SE(3) irreps. These merged 6D point clouds are then processed through novel SE(3)xSE(3)-transformer layers derived from the Wigner-Eckart theorem of G-steerable kernels [5,17], and finally through an SE(3) projection layer that resembles Arun’s method.

The paper also includes theoretical analysis showing that BITR effectively inherits the symmetry properties of the classical Arun’s method in three aspects: 1) SE(3)-bi-equivariance, 2) Swap equivariance, and 3) Scale equivariance. This enables effective point cloud assembly without assuming known point correspondence. These benefits are also validated through extensive comparison with state-of-the-art baselines.

**Strengths:**

1. To the best of the reviewers' knowledge, this is the first method to achieve SE(3)-bi-equivariance in every layer. While several existing works in biology [1] and robotics [2,3] have addressed SE(3)-bi-equivariance, they only managed it in an ad hoc, template-matching style in the final layer. In contrast, the proposed $SE(3)\times SE(3)$-transformer layer is inherently bi-equivariant, theoretically grounded in the Wigner-Eckart theorem, and thus expected to be more general and expressive than previous approaches.

2. Experimental results show that the proposed BITR significantly improves point cloud assembly accuracy compared to previous methods. In particular, BITR retains accuracy when the overlap is small, whereas other methods do not.

3. Although not included in the main sections, U-BITR presented in Appendix C is also innovative and promising

4. The proposed method has the potential for broader applications beyond point cloud assembly. It could be beneficial in various areas that require inferring the relative SE(3) pose between two point clouds (or graphs), such as protein docking, robotic manipulation, and camera pose estimation.

Overall, this paper presents a novel and theoretically solid approach. I am confident that it is a breakthrough in bi-equivariant modeling for SE(3) pose inference problems, including point cloud assembly tasks. The empirical results are sufficient to support the claimed benefits.

[1] Ganea et al., "Independent SE(3)-equivariant Models for End-to-end Rigid Protein Docking,” ICLR 2022

[2] Ryu et al., “Diffusion-EDFs: Bi-equivariant Denoising Generative Modeling on SE(3) for Visual Robotic Manipulation,” CVPR 2024

[3] Huang et al., “Fourier Transporter: Bi-Equivariant Robotic Manipulation in 3D,” ICLR 2024

**Weaknesses:**

1. The paper may be challenging for readers without a background in representation theory and SE(3)-equivariant neural networks. The proposed method is complicated and might be difficult to implement and extend.

2. ~~Experimental validation is rather confined to simple problems in which object parts are clipped by a random plane. It isn’t sufficiently verified for more general and realistic assembly tasks in which the parts are not cleanly cut by a plane.~~ Addressed.

**Questions:**

1. Why include type-1 features in the last layer before softmax in Eq (12)? Why not use only type-0 irreps? Why not use type-2 or higher?

2. How is locality (neighborhood) defined for 6D point clouds? Directly using KNN for 6D point clouds to construct a neighborhood graph does not make sense to me.

3. The key points of X and Y are ordered and thus not permutation-invariant. Could this have any negative consequences?

**Limitations:**

1. The proposed method is not generative and thus cannot handle scenarios where multiple assembly poses are equally valid.

2. The $SE(3)\times SE(3)$-transformer layer in BITR relies on the Clebsch-Gordon tensor product, which is notorious for its computational complexity, scaling with $O(L^6)$ where $L$ is the maximum degree. The paper proposes a 2nd-order Clebsch-Gordon tensor product, which likely has similar or worse complexity. The lack of efficient CUDA kernels exacerbates this problem.

These limitations are appropriately discussed by the authors in Appendix E. The reviewer believes that these limitations are not fundamental and could be addressed in future research. For example, recently proposed eSCN Convolution [4] and Gaunt tensor product [5] have reduced the computational complexity of the C-G tensor product from $O(L^6)$ to $O(L^3)$. Extending these state-of-the-art $SE(3)$-equivariant mechanisms into a bi-equivariant form would be an interesting future direction.

[4] Passaro et al., "Reducing SO(3) Convolutions to SO(2) for Efficient Equivariant GNNs," ICML 2023

[5] Luo et al., "Enabling Efficient Equivariant Operations in the Fourier Basis via Gaunt Tensor Products," ICLR 2024

---

> ### Author Rebuttal · Authors · 2024-08-05
>
> Thanks for the careful reading of our paper.
> We are happy you like the extended U-BITR model which is not even included in the main text due to space limitations.
> Thanks for bringing up [4,5], we have read through these papers, and they indeed seem useful for accelerating our method.
> We now address your concerns as follows.
>
> 1. The paper may be challenging for readers without a background in representation theory and SE(3)-equivariant neural networks. The proposed method is complicated and might be difficult to implement and extend.
>
> We agree that some non-trivial preliminaries are required to read this paper. To alleviate this, we included a brief introduction of SE3-transformers in Appx.A to make the material self-contained. We also (briefly) introduced representation theory in Sec.3.2. [5] is cited and readers can find more complete information there. As for the implementation, we will release the code upon accept to facilitate the research in this area.
>
> 2. Experimental validation is rather confined to simple problems in which object parts are clipped by a random plane. It isn’t sufficiently verified for more general and realistic assembly tasks in which the parts are not cleanly cut by a plane.
>
> We have tested our method on datasets which are not generated by cutting. For example, the BB, 7Scene and ASL datasets from Sec 6.4 to Sec. 6.6, where BB is generated by physics simulation and the other two are real datasets. Also, the data in the manipulation experiment is not generated by cutting.
>
>
> 3. Why include type-1 features in the last layer before softmax in Eq (12)? Why not use only type-0 irreps? Why not use type-2 or higher?
>
> We only use degree-$\\{0, 1\\}$ features in our network due to efficiency considerations. In the merging layer, we merge these features. We will add a sentence below Eqn 12 to make this more clear: "here we only include degree-0 and degree-1 features, but higher degree features can be used similarly"
>
> 4. How is locality (neighborhood) defined for 6D point clouds? Directly using KNN for 6D point clouds to construct a neighborhood graph does not make sense to me.
>
> We use KNN in 6D space. Our rationale is that a small distance in 6D space means small distances in the first and last 3D components ($\tilde{X}$ and $\tilde{Y}$), i.e. the key point coordinates of X and Y. So KNN in 6D is similar to doing KNN for $\tilde{X}$ and $\tilde{Y}$ and then taking the intersection of the edges of these two graphs, and it has the advantage of avoiding empty edges for all points.
>
>
>
> 5. The key points of X and Y are ordered and thus not permutation-invariant. Could this have any negative consequences?
>
> No, they are permutation invariant. The coordinate of each key point $k_j$ is computed as $k_j = \sum_{i} F_{ji}X_i$, where $F_{ji}$ is an invariant feature at node $i$ (after softmax normalization). So if we permute $i$, we still get the same $k_j$  for all $j$.

---

> > ### Comment · Reviewer_fq5D · 2024-08-13
> >
> > Thanks for the clarification. All concerns and questions have been addressed.

---

### Official Review · Reviewer_mtoz · 2024-07-12

**Soundness:** 3
**Presentation:** 2
**Contribution:** 3
**Rating:** 5
**Confidence:** 3

**Summary:**

The paper addresses the problem of assembling point clouds. Given two partial, potentially unmatched point clouds, the objective is to determine the rigid transformation that best aligns them in relation to the unknown complete shape. The proposed approach is a learning-based method, utilizing an architecture that adheres to symmetry constraints inspired by Arun's method. The effectiveness of the method is demonstrated through a toy experiment and synthetic data, with evaluations on real data included in the appendix.

**Strengths:**

The paper tackles a challenging task.

Deriving the required properties of the suggested network from Arun's method is an elegant solution. The theoretical analysis supporting the suggested method seems to be solid.

The provided toy experiment seems to be convincing.

**Weaknesses:**

The presentation quality could be improved. For example, figure 2 is hard to understand. The description of the toy experiment is not easy to follow as well.

Evaluating the benefits of bi-equivariance. An alternative to bi-equivariance, is to train a regular equivarince network on the task of shape completion and then align the two partial inputs according to the predicted completed shape (using one of the baselines that use correspondences).

**Questions:**

Does the suggsted bi-equivariance architecture generalize better than a single-equivariance network trained on shape completion?

**Limitations:**

yes

---

> ### Author Rebuttal · Authors · 2024-08-05
>
> We thank the reviewer for the time and effort. We address the concerns below.
>
> 1. The presentation quality could be improved. For example, figure 2 ... The description of the toy experiment..
>
> We apologize for the lack of clarity. We kept all descriptions concise to fit in the limited 9-page constraint. However, we have now added more descriptions to the caption of Fig 2 and revised the description of the toy example. We have now cited [37] to provide more visualization of the generated data.
>
> We have made the following modification (the added content is marked bold):
> - (The caption of Fig.2)  "The input 3-D PCs X and Y are first merged into a 6-D PC Z **by concatenating the extracted key points $\tilde{X}$ and $\tilde{Y}$"**....
> - (Line 298)..We train BITR on the bunny shape. **We prepare the dataset similar to [37]:** In each training iteration.....We train BITR to ~~reconstruct S using~~ **randomly rotated and translated** $\\{X_P, Y_P\\}$...
>
> 2. Evaluating the benefits of bi-equivariance. An alternative to bi-equivariance, is to train a regular equivariance network on the task of shape completion and then align the two partial inputs according to the predicted completed shape.....  Does the suggested bi-equivariance architecture generalize better than a single-equivariance network trained on shape completion?
>
> We have evaluated the benefits of bi-equivariance against non-equivariant methods [21] and SE(3)-equivariant method [35]. We observed some improvement against these methods in our experiments.
>
> On the other hand, we are not aware of any assembly/registration method that is based on shape completion, nor do we think this can be implemented easily. Because the two generated completed shapes may not be consistent (there is no correspondence between the generated part), unless both of the inputs are taken into consideration in the generation. But that will require assembling the inputs first, or at least using SE(3)-bi-equivariant features, which falls back to the assembly task. In summary, we do not think point cloud completion based methods can be used in this task.

---

> > ### Comment · Reviewer_mtoz · 2024-08-12
> > **reply to authors**
> >
> > I thank the authors for their rebuttal. However, I remain concerned about the benefits of bi-equivariance compared to simpler baselines. To reiterate my suggestion: would it be possible to use an equivariant network trained on shape completion, and then apply a simple method like ICP to register the equivariant network outputs for the assembly task?

---

> > > ### Author Response · Authors · 2024-08-12
> > >
> > > Thanks for your reply.
> > >
> > > As we mentioned in our first reply. No. We do not think the suggested method will work, nor do we know any assembly method like the suggested one. Because the generated complete shapes do not necessarily have corresponding points due to the non-uniqueness of shape completion (even for high quality completions). For example, for the outdoor scene (like the one in Fig 11), let's say a tree is observed in $Y$ but not in $X$. It is difficult to generate a tree to complete $X$ and guarantee that this tree is exactly the one observed in $Y$. (It would be difficult to guarantee that a tree is generated (the algorithm may generate a house instead of a tree) and this generated tree is exactly the one in $Y$ (the same position,  shape and orientation). )  This ambiguity also exists for other type of data like indoor scenes (a different table in a room), objects (a different tail of the airplane in Fig 1b), which makes the generated shape not suitable for the assembly task.

---

> > > ### Comment · Reviewer_fq5D · 2024-08-13
> > > **Opinion on the shape completion**
> > >
> > > I don't believe shape completion can replace bi-equivariance because there is always some uncertainty in shape completion. To show this, consider two parts, $X$ and $Y$. If one infers $g$ from the reconstruction $\hat{Y}\sim P(\hat{Y}|X)$, the marginal distribution of the inferred pose is $P(g|X,Y)=\int P(g|\hat{Y},Y)P(\hat{Y}|X) d\hat{Y}$, whose integral is intractable unless $P(\hat{Y}|X)$ is deterministic. In contrast, bi-equivariant models, which do not rely on shape completion, can directly infer $P(g|X,Y)$. Although BITR is not a generative model, the argument still holds that it does not require intractable marginalization over the uncertain shape completion $\hat{Y}$.

---

### Official Review · Reviewer_P5ME · 2024-07-30

**Soundness:** 2
**Presentation:** 3
**Contribution:** 2
**Rating:** 4
**Confidence:** 5

**Summary:**

This paper proposes a novel architecture that extends the SE(3) transformers to become bi-equivariant for the task of rigid point cloud assembly. That is the authors enforce a single assembly for all rigid transformations of the source or target point clouds that is learnable by the network. They also propose how to extend the method for scale and swap consistent assembly of the point clouds. The authors aim to design a method that is correspondence-free to deal with the cases where the point clouds have zero overlap. They first learn (the same number of) keypoints on the original point clouds and concatenate them channel wise. After extracting the bi-equivariant features they utilize the Arun's algorithm to align the point clouds. Some experiments in small-scale datasets of zero overlap showcase benefits of the method.

**Strengths:**

- The authors aim to address a significant problem of pairwise PC registration when the PC parts have close to zero / zero overlap.
- The derivations of the layer parametrization seem correct (although missing details on second order CG coefficients).
- The experiments show some performance gain in certain cases.

**Weaknesses:**

- **On the assumptions/setup** : The setup of zero overlap is not addressed properly when equivariant constraints are enfroced. In the zero overlap case multiple correct poses is inherent. For example a chair next to a table can be placed in many "correct" ways. Some of them will be in the training data. But when two different configurations are presented in the training data that breaks the equivariant assumption. Or symmetrically, enforcing equivariance from the first configuration can never reduce the error of the second configuration. I believe the problem of zero overlap has to be formulated more carefully and these consequences should be discussed.
- **On the method**:
    1. **Correspondence-free methods should take care of the permutations of the points**: Concatenating the two point clouds after processing them independently does not properly take care of the permutation of points. In fact an individual permutation of the points in X,Y would result in permuted keypoints in Eq.(10). When the keypoints are concatenated each keypoint will match a different keypoint which means that the bi-equivariant transformer will view the permuted case as completely different features.
    2. **Assumptions on the sizes of the parts**: Extracting the same amount of keypoints from both point clouds might be restrictive when sizes differ by a lot.

- **On the experiments**:  Fewer experiments are performed in this paper than what is standard in the literature. Also the scale of the datasets is much smaller in number and sizes.
   - Low-overlap settings (<10%) exist in KITTI, 3DMatch  that the authors do not consider. This also raises the question of the scalability of the method.
   - The sizes of the parts also answer the question regarding extracting fixed number of keypoints from the two point clouds. E.g. for large point clouds as in 3DMatch how many keypoints are enough?

- **References**:  Missing literature. A lot of point cloud processing papers utilize the SE(3) transformer (some are even bi-equivariant) for registration, reconstruction, docking etc. Others address equivariant registration with other networks. Some suggestions:
    - On Point Cloud Registration:
       - M.Zhu et al. "Correspondence-Free Point Cloud Registration with SO(3)-Equivariant Implicit Shape Representations."
       - C.Lin et al  "Coarse-to-Fine Point Cloud Registration with SE(3)-Equivariant Representations."
    - On PC Processing using SE(3)-transformer and/or bi-equivariance:
        - E. Chatzipantazis et al.   "SE(3)-Equivariant Attention Networks for Shape Reconstruction in Function Space".
        - Y.Peng et al "SE(3)-Diffusion: An Equivariant Diffusion Model for 3D Point Cloud Generation."
        - O. Ganea et al. "INDEPENDENT SE(3)-EQUIVARIANT MODELS FOR END-TO-END RIGID PROTEIN DOCKING."
        - C.Lin et al. "SE(3)-Equivariant Point Cloud-Based Place Recognition".

**Questions:**

- Line 28: "Sensitive to the initial poses of PCs". Is there any reference in the literature for that?
- In practice we do not know id the point clouds have zero overlap or some overlap. How can you discriminate such cases?
- The zero overlap case could in principle involve many solutions. For example a chair next to a table. Enforcing the configuration that the training data suggest and restricting it with equivariant constraints can hurt performance as the dataset could have other possible configurations for other training samples that are inconsistent with the constraint.
- The setup of zero overlap has to be formulated more properly.
- In the method the authors extract the same amount of keypoints for the X,Y point clouds. Is that restricitve in cases one point cloud is larger than the other?
- Shouldn't there be some sorting of the keypoints before concatenation? See Weaknesses too.
- Why is BITR+ICP bi-equivariant?
- How does the method scale to large point clouds with complicated distributions as in 3DMatch?
- Are normals given as features in this and the rest of the methods? Are the normals computed before or after the cut?
- Is the method trained per-class or for the whole dataset?

**Limitations:**

See the section in Weaknesses and then Questions.

---

> ### Author Rebuttal · Authors · 2024-08-05
>
> We thank the reviewer for the time and effort. We address the concerns below.
>
> 1. The setup of zero overlap is not addressed properly when equivariant constraints are enforced...
>
> The uniqueness and overlap ratio are completely different concepts. There can be non-unique assembly even in the fully overlapped case. Consider, for example, aligning a sphere to another sphere of the same size. The uniqueness is already discussed in Appx.E.
>
>
> 2. ...permutation of the points in X,Y would result in permuted keypoints in Eq.(10).
>
> This is not true. Eqn 10 is a quite common technique to get invariant key points, see the definition of $y_{1k}$ and $y_{2k}$ in Independent SE(3)-equivariant Models for End-to-end Rigid Protein Docking,
> which is a paper mentioned by the reviewer.
>
> 3. Extracting the same amount of keypoints from both point clouds might be restrictive when sizes differ by a lot.
>
> We do not see any restriction. If the reviewer thinks there is a restriction, evidence should be provided.
>
>
> 4. standard in the literature. Also the scale of the datasets is much smaller in number and sizes.
>
> Our method is an assembly method where the input is not necessarily overlapped, not just a registration task for overlapped input. Those datasets might be standard for registration, but not for assembly.
> If the reviewer thinks the scalability of our method is an issue, then we insist a more scalable baseline method on this task must be provided.
>
>
> 5. References
>
> We have now added citations
> "Coarse-to-Fine Point Cloud Registration with SE(3)-Equivariant Representations" and "Independent SE(3)-equivariant Models for End-to-end Rigid Protein Docking" in line 88 (..modelling 3D data []).
> We will not cite other papers because they are not related to the subject.
>
>
>
> 6. Line 28: "Sensitive to the initial poses of PCs". Is there any reference in the literature for that?
>
> It's a well-known fact. For example, the last sentence of the 2nd paragraph in [36] "All these methods are sensitive to the initial positions."
>
> 7. In practice we do not know id the point clouds have zero overlap or some overlap. How can you discriminate such cases?
>
> We don't think there is an easy way to discriminate this. That's actually an important reason why our method is more flexible than registration methods:
> the registration methods will fail silently when the point clouds are not overlapped.
>
> 8 and 9. The zero overlap case could in principle involve many solutions.
>
> See 1.
>
>
> 10. In the method the authors extract the same amount of keypoints for the X,Y point clouds. Is that restrictive in cases one point cloud is larger than the other?
>
> See 3
>
>
> 10. Shouldn't there be some sorting of the keypoints before concatenation? See Weaknesses too.
>
> See 2.
>
> 11. Why is BITR+ICP bi-equivariant?
>
> Because ICP is distance-based.
>
>
> 12. How does the method scale to large point clouds with complicated distributions as in 3DMatch?
>
> For a "complicated" indoor dataset, see experiments on 7Scenes. See Appx E for computational cost.
>
>
> 13. Are normals given as features in this and the rest of the methods? Are the normals computed before or after the cut?
>
> 1. Normals are use in ROI, not other mehtods. 2. After.
>
> 14. Is the method trained per-class or for the whole dataset?
>
> per-class

---

> > ### Comment · Reviewer_P5ME · 2024-08-12
> > **Response to Authors**
> >
> > ## Permutation Equivariance:
> >
> > I believe there is methodological error in this paper and the authors have not answered correctly the **permutation equivariance** question that has been raised by many reviewers. I will explain the problem next because I believe it will help the authors fix it in the future:
> >
> > Concatenating the keypoints $\tilde{X}, \tilde{Y}$ that are extracted using equation (10) means that if the point clouds are presented with a different numbering of the points i.e. $P_1X, P_2Y$ for $P_1,P_2$ permutation matrices (which is **always the case** for point clouds), then the keypoints are also permuted with different permutations ie. $P_1 \tilde{X}, P_2 \tilde{Y}$. (They are not invariant as the authors suggest in their answer to reviewer fq5D because the index/node $j$ moves too to some $j'$ not only the index of the neighbors $i$; in other words, the **same feature** appears but **in different position in the vector**). Then, when the features are concatenated one gets $z_u =  x_{u'} \oplus y_{u''}$  for different $u',u''$ that depend on $P_1,P_2$. Thus, **the same input point clouds** will produce **different vectors $z$**! This breaks permutation equivariance and potentially needs many augmentations to recover. Moreover, it is not clear what this vector $z$ now consists of since all combinations are possible.
> >
> > This is not the same (as the authors suggested) for "INDEPENDENT SE(3)-EQUIVARIANT MODELS FOR END-TO-END RIGID PROTEIN DOCKING" because in that paper the keypoints are enforced to correspond by the optimal transport loss. The authors do not do such a thing (and maybe cannot, because it would turn the method from a correspondence-free to a correspondence based).
> >
> > This is probably why the method cannot generalize beyond per-class settings (like registration methods do) and even require many cuts from a single point cloud to do the assembly (overfitting).
> >
> > ## Non-uniqueness and overlap:
> >
> > This question I thing is not answered correct either. Of course, even in the large overlap setting there can be many solutions (mainly due to symmetry). The argument was not that though. In the no-overlap setting it is very common that there might be multiple solutions (even if no symmetric objects exist). For example, if one part is a table and another is a chair (that is we have no overlap) then the geometric relation turns into a functional one. You can put the chair near many places in the table. That is not due to the symmetry of the parts. That is why I am arguing that the setting need a better formulation regarding uniqueness. If in  the data there are tables and chairs in different configurations satisfying both the data and the equivariant loss is impossible. This has not been properly discussed in the limitations.
> >
> > ## Same amount of keypoints in two parts:
> >
> > The authors have not answered this question either which I think is important especially in cases that the two point cloud parts differ by a lot. In order to observe that fact, the paper needs to provide more strong experimental results than toy datasets like per-class ShapeNet.
> >
> > ## Comment on the weak experimentation:
> >
> > The authors after the questions admit that the non-overlap setting cannot be identified. Thus it is only reasonable that the method should perform equally well to a setting of small overlap (or when no-overlap and small overlap exist in the dataset). The authors also claim that the method is not "just a registration method" in the answer above which implies that it will not fail silently in such situations where small overlap exists. However, no experimentation in standard datasets with low overlap (<10%) like 3DLoMatch have been provided. I argued above why this might be a methodological issue. It is also probably a memory scalability issue.
> >
> > Even in the 7Scenes  dataset, which I believe should be in the main text,  the method does not outperform GeoTransformer or ROI even when the authors replicate their steps to make their method as similar as possible to the other papers. Also it is not compared properly against other state-of the-art methods (like Lepard: Learning partial point cloud matching in rigid and deformable scenes, You Only Hypothesize Once: Point Cloud Registration with Rotation-equivariant Descriptors
> > ).
> >
> > In total, based on the arguments above and the responses of the authors I believe that the paper has a good approach to the problem, however, its major methodological errors in the design and weak experimentation cannot reveal the benefits of the method in non-toy settings. I cannot in good conscience suggest acceptance of the paper as is, but I believe that if the authors take the reviews into consideration it could potentially improve the approach a lot.

---

> ### Author Response · Authors · 2024-08-12
>
> Thanks for your reply.  It is quite easy to see that the key points are permutation invariant, simply because the set of features of neiborhood points are always the same under permutation. (It has nothing to do with optimal transport) In other words, the whole network does not rely on the specific index of the node.

---

### Decision · Program_Chairs · 2024-09-25

**Decision:**

Accept (poster)

**Comment:**

The paper introduces a bi-equivariant transformer model to solve the task of pointcloud assembly, where two pointclouds with no overlapping regions need to be registered. Although most of the reviewers provided an initial positive assessment of the work, one reviewer recommended rejecting the paper on the grounds that the method is not equivariant. After a careful read of the paper and authors' response, we concluded that this is due to a misunderstanding of Eq.10 of the paper by the reviewer and the method is in fact equivariant. Therefore, although several reviewers raised the concern of an evaluation focused only on small datasets or missing baselines, we will follow the recommendation of the other reviewers and recommend the paper for acceptance. However, in the final version of the paper, the authors should include the feedback received from reviewers and improve clarity in the method's description (especially regarding the equivariant nature of the method and why Eq.10 makes the method equivariant) and the experiments section.